# Robust low-rank training via approximate orthonormal constraints

**Dayana Savostianova**
Gran Sasso Science Institute
67100 L'Aquila (Italy)
`dayana.savostianova@gssi.it`

**Emanuele Zangrando**
Gran Sasso Science Institute
67100 L'Aquila (Italy)
`emanuele.zangrando@gssi.it`

**Gianluca Ceruti**
University of Innsbruck
6020 Innsbruck (Austria)
`gianluca.ceruti@uibk.ac.at`

**Francesco Tudisco**
Gran Sasso Science Institute
67100 L'Aquila (Italy)
`francesco.tudisco@gssi.it`

## Abstract

With the growth of model and data sizes, a broad effort has been made to design pruning techniques that reduce the resource demand of deep learning pipelines, while retaining model performance. In order to reduce both inference and training costs, a prominent line of work uses low-rank matrix factorizations to represent the network weights. Although able to retain accuracy, we observe that low-rank methods tend to compromise model robustness against adversarial perturbations. By modeling robustness in terms of the condition number of the neural network, we argue that this loss of robustness is due to the exploding singular values of the low-rank weight matrices. Thus, we introduce a robust low-rank training algorithm that maintains the network's weights on the low-rank matrix manifold while simultaneously enforcing approximate orthonormal constraints. The resulting model reduces both training and inference costs while ensuring well-conditioning and thus better adversarial robustness, without compromising model accuracy. This is shown by extensive numerical evidence and by our main approximation theorem that shows the computed robust low-rank network well-approximates the ideal full model, provided a highly performing low-rank sub-network exists.

## 1   Introduction

Deep learning and neural networks have achieved great success in a variety of applications in computer vision, signal processing, and scientific computing, to name a few. However, their robustness with respect to perturbations of the input data may considerably impact security and trustworthiness and poses a major drawback to their real-world application. Moreover, the memory and computational requirements for both training and inferring phases render them impractical in application settings with limited resources. While a broad literature on pruning methods and adversarial robustness has been developed to address these two issues in isolation, much less has been done to design neural networks that are both energy-saving and robust. Actually, in many approaches the two problems seem to compete against each other as most adversarial robustness improving-techniques require even larger networks [36, 41, 46, 47, 79] or computationally more demanding loss functions, and thus more expensive training phases [13, 23, 34, 44, 66].

The limited work available so far on robust pruned networks is mostly focused on reducing memory and computational costs of the inference phase, while retaining adversarial robustness [20, 28, 42, 56, 72, 76]. However, the inference phase amounts to only a very limited fraction of the cost of the

37th Conference on Neural Information Processing Systems (NeurIPS 2023).

whole deep learning pipeline, which is instead largely dominated by the training phase. Reducing both inference and training costs is a challenging but desirable goal, especially in view of a more accessible AI and its effective use on limited-resource and limited-connectivity devices such as drones or satellites.

Some of the most effective techniques for the reduction of training costs so far have been based on low-rank weights parametrizations [29, 55, 71]. These methods exploit the intrinsic low-rank structure of parameter matrices and large data matrices in general [17, 49, 59, 69]. Thus, assuming a low-rank structure for the neural network's weights $W = USV^\top$, the resulting training procedures only use the small individual factors $U, S, V$. This results in a training cost that scales linearly with the number of neurons, as opposed to a quadratic scaling required by training full-rank weights. Despite significantly reducing training parameters, these methods achieve accuracy comparable with the original full networks. However, their robustness with respect to adversarial perturbations has been largely unexplored so far.

**Contributions**

In this paper, we observe that the adversarial robustness of low-rank networks may actually deteriorate with respect to the full baseline. By modeling the robustness of the network in terms of the neural network's condition number, we argue that this loss of robustness is due to the exploding condition number of the low-rank weight matrices, whose singular values grow very large in order to match the baseline accuracy and to compensate for the lack of parameters. Thus, to mitigate this growing instability, we design an algorithm that trains the network using only the low-rank factors $U, S, V$ while simultaneously ensuring the condition number of the network remains small. To this end, we interpret the loss optimization problem as a continuous-time gradient flow and use techniques from geometric integration theory on manifolds [12, 31, 55, 70] to derive three separate projected gradient flows for $U, S, V$, individually, which ensure the condition number of the network remains bounded to a desired tolerance $1 + \tau$, throughout the epochs. For a fixed small constant $\varepsilon > 0$, this is done by bounding the singular values of the small rank matrices within a narrow band $[s - \varepsilon, s + \varepsilon]$ around a value $s$, chosen to best approximate the original singular values.

We provide several experimental evaluations on different architectures and datasets, where the robust low-rank networks are compared against a variety of baselines. The results show that the proposed technique allows us to compute from scratch low-rank weights with bounded singular values, significantly reducing the memory demand and computational cost of training while at the same time retaining or improving both the accuracy and the robust accuracy of the original model. On top of the experimental evidence, we provide a key approximation theorem that shows that if a high-performing low-rank network with bounded singular values exists, then our algorithm computes it up to a first-order approximation error.

This paper focuses on feed-forward neural networks. However, our techniques and analysis apply straightforwardly to convolutional filters reshaped in matrix form, as done in e.g. [29, 55, 71]. Other ways exist to promote orthogonality of convolutional filters, e.g. [60, 65, 77], which we do not consider in this work.

## 2   Related work

Neural networks' robustness against adversarial perturbations has been extensively studied in the machine learning community. It is well-known that the adversarial robustness of a neural network is closely related to its Lipschitz continuity [13, 23, 62, 68], see also Section 3. Accordingly, training neural networks with bounded Lipschitz constant is a widely employed strategy to address the problem. A variety of works studied Lipschitz architectures [37, 60, 65, 68], and a number of certified robustness guarantees have been proposed [23, 51, 61]. While scaling each layer to impose 1-Lipschitz constraints is a possibility, this approach may lead to vanishing gradients and it is known that a more effective way to reduce the Lipschitz constant and increase robustness is obtained by promoting orthogonality on each layer [5, 13]. On top of robustness, small Lipschitz constants and orthogonal layers are known to lead to improved generalization bounds [11, 45] and more interpretable gradients [67]. Orthogonality was also shown to improve signal propagation in (very) deep networks [52, 74].

A variety of methods to integrate orthogonal constraints in deep neural networks have been developed over the years. Notable example approaches include methods based on regularization and landing [1, 13], cheap parametrizations of the orthogonal group [6, 38, 39, 48, 50], Riemannian and projected gradient descent schemes [2, 3, 10].

In parallel to the development of methods to promote orthogonality, an active line of research has grown to develop effective training strategies to enforce low-rank weights. Unlike sparsity-promoting pruning strategies that primarily aim at reducing the parameters required for inference [8, 21, 22, 30, 43], low-rank neural network models are designed to train directly on the low-parametric manifold of low-rank matrices and are particularly effective to reduce the number of parameters required by both inference and training phases. Similar to orthogonal training, methods for low-rank training include methods based on regularization [26, 29], as well as methods based on efficient parametrizations of the low-rank manifold using the SVD, randomized tensor dropout or the polar decomposition [32, 71, 75], and Riemannian optimization-based training models [55, 57].

By combining low-rank training with approximate orthogonal constraints, in this work we propose a strategy that simultaneously enforces robustness while only requiring a reduced percentage of the network's parameters during training. The method is based on a gradient flow differential formulation of the training problem, and the use of geometric integration theory to derive the governing equations of the low-rank factors. With this formulation, we are able to reduce the sensitivity of the network during training at almost no cost, yielding well-conditioned low-rank neural networks. Our experimental findings are supported by an approximation theorem that shows that, if the ideal full network can be approximated by a low-rank one, then our method computes a good approximation. This is well-aligned with recent work that shows the existence of high-performing low-rank nets in e.g. deep linear models [7, 17, 25, 49]. Moreover, as orthogonality helps in training really deep networks, low-rank orthogonal models may be used to mitigate the effect of increased effective depth when training low-rank networks [54].

## 3 The condition number of a neural network

The adversarial robustness of a neural network model $f$ can be measured by the worst-case sensitivity of $f$ with respect to small perturbations of the input data $x$. In an absolute sense, this boils down to measuring the best global and local Lipschitz constant of $f$ with respect to suitable distances, as discussed in a variety of papers [13, 15, 23, 62]. However, as the model and the data may assume arbitrary large and arbitrary small values in general, a relative measure of the sensitivity of $f$ may be more informative. In other words, if we assume a perturbation $\delta$ of small size as compared to $x$, we would like to quantify the largest relative change in $f(x + \delta)$, as compared to $f(x)$. This is a well-known problem of conditioning, as we review next, and naturally leads to the concept of condition number of a neural network.

In the linear setting, the condition number of a matrix is a widely adopted relative measure of the worst-case sensitivity of linear problems with respect to noise in the data. For a matrix $A$ and the matrix operator norm $\|A\| = \sup_{x \neq 0} \|Ax\|/\|x\|$ the condition number of $A$ is defined as $\mathrm{cond}(A) = \|A\|\|A^+\|$, where $A^+$ denotes the pseudo-inverse of $A$. Note that it is immediate to verify that $\mathrm{cond}(A) \geq 1$. Now, if for example $u$ and $u_\varepsilon$ are the solutions to the linear system $Au = b$, when $A$ and $b$ are exact data or when they are perturbed with noise $\delta_A$, $\delta_b$ of relative norm $\|\delta_A\|/\|A\| \leq \varepsilon$ and $\|\delta_b\|/\|b\| \leq \varepsilon$, respectively, then the following relative error bound holds

$$\frac{\|u - u_\varepsilon\|}{\|u\|} \lesssim \mathrm{cond}(A)\, \varepsilon \,.$$

Thus, small perturbations in the data $A, b$ imply small alterations in the solution if and only if $A$ is well conditioned, i.e. $\mathrm{cond}(A)$ is close to one.

As in the linear case, it is possible to define the concept of condition number for general functions $f$, [24, 53]. Let us start by defining the relative error ratio of a function $f : \mathbb{R}^d \to \mathbb{R}^m$ in the point $x$:

$$R(f, x; \delta) = \frac{\|f(x + \delta) - f(x)\|}{\|f(x)\|} \bigg/ \frac{\|\delta\|}{\|x\|}. \tag{1}$$

In order to take into account the worst-case scenario, the *local* condition number of $f$ at $x$ is defined by taking the sup of (1) over all perturbations of relative size $\varepsilon$, i.e. such that $\|\delta\| \leq \varepsilon \|x\|$, in the

limit of small $\varepsilon$. Namely, $\mathrm{cond}(f;x) = \lim_{\varepsilon\downarrow 0} \sup_{\delta\neq 0: \|\delta\|\leq\varepsilon\|x\|} R(f,x;\delta)$. This quantity is a local measure of the "infinitesimal" conditioning of $f$ around the point $x$. In fact, a direct computation reveals that

$$\frac{\|f(x+\delta) - f(x)\|}{\|f(x)\|} \lesssim \mathrm{cond}(f;x)\,\varepsilon\,, \qquad (2)$$

as long as $\|\delta\| \leq \varepsilon\|x\|$. Thus, $\mathrm{cond}(f;x)$ provides a form of relative local Lipschitz constant for $f$ which in particular shows that, if $\|\delta\|/\|x\|$ is smaller than $\mathrm{cond}(f;x)^{-1}$, we expect limited change in $f$ when $x$ is perturbed with $\delta$. A similar conclusion is obtained using an absolute local Lipschitz constant in e.g. [23]. Similarly to the absolute case, a global relative Lipschitz constant can be obtained by looking at the worst-case over $x$, setting $\mathrm{cond}(f) = \sup_{x\in\mathcal{X}} \mathrm{cond}(f;x)$. Clearly, the same bound (2) holds for $\mathrm{cond}(f)$. Note that this effectively generalizes the linear case, as when $f(x) = Ax$ we have $\mathrm{cond}(f) = \mathrm{cond}(f,x) = \mathrm{cond}(A)$.

When $f$ is a neural network, $\mathrm{cond}(f)$ is a function of the network's weights and robustness may be enforced by reducing $\mathrm{cond}(f)$ while training. In fact, $\mathrm{cond}(f)$ is the relative equivalent of the network's Lipschitz constant and thus standard Lipschitz-based robustness certificates [23, 40, 68] can be recast in terms of $\mathrm{cond}(f)$. However, for general functions $f$ and general norms $\|\cdot\|$, $\mathrm{cond}(f)$ may be (very) expensive to compute, it may be non-differentiable, and $\mathrm{cond}(f) > 1$ can hold [24]. Fortunately, for feed-forward neural networks, it holds (proof and additional details moved to Appendix B in the supplementary material)

**Proposition 1.** *Let $\mathcal{X}$ be the input space and let $f(x) = z_{L+1}$ be a network with $L$ linear layers $z_{i+1} = \sigma_i(W_i z_i)$, $i = 1,\ldots,L$. Then,*

$$\mathrm{cond}(f) = \sup_{x\in\mathcal{X}\setminus\{0\}} \mathrm{cond}(f;x) \leq \Big(\prod_{i=1}^{L} \sup_{x\in\mathcal{X}_i\setminus\{0\}} \mathrm{cond}(\sigma_i;x)\Big)\Big(\prod_{i=1}^{L} \mathrm{cond}(W_i)\Big).$$

*In particular, for typical $\mathcal{X}_i$ and typical choices of $\sigma_i$, including $\sigma_i \in \{$leakyReLU, sigmoid, tanh, hardtanh, softplus, siLU$\}$, we have*

$$\sup_{x\in\mathcal{X}_i\setminus\{0\}} \mathrm{cond}(\sigma_i;x) \leq C < +\infty$$

*for a positive constant $C > 0$ that depends only on the activation function $\sigma_i$.*

Note that for entrywise nonlinearities $\sigma$, the condition number $\mathrm{cond}(\sigma;x)$ can be computed straightforwardly. In fact, when $\sigma$ is Lipschitz, the problem can be reduced to a one-dimensional function, and it follows directly from its definition that (see also [64])

$$\mathrm{cond}(f;x) = \sup_{\nu_x\in\partial\sigma(x)} |\nu_x||x||\sigma(x)|^{-1}, \qquad x\in\mathbb{R}$$

where $\partial\sigma(x)$ denotes Clarke's generalized gradient [14] of $\sigma$ at the point $x$. Thus, for example, if $\sigma$ is *LeakyRelu* with slope $\alpha$, we have $\mathrm{cond}(\sigma) = 1$; if $\sigma$ is the *logistic sigmoid* $(1 + e^{-x})^{-1}$ and the feature space $\mathcal{X}_i$ is such that $\mathcal{X}_i = W_i\mathcal{X}_{i-1}$, then if $|z_{i-1}| \leq c_{i-1}$ entry-wise, we have $|x_i| \leq c_{i-1}\max_{uv} |W_i|_{uv} := c_i$ and $\mathrm{cond}(\sigma) \leq \sup_{x\geq -c_i} |x|e^{-x}(1 + e^{-x})^{-1} \leq \max\{c_i, 1/e\}$.

From Proposition 1 we see that when $f$ is a feed-forward network, to reduce the condition number of $f$ it is enough to reduce the conditioning of all its weights. When $\|\cdot\| = \|\cdot\|_2$ is the Euclidean $L^2$ norm, we have $\mathrm{cond}_2(W) = s_{\max}(W)/s_{\min}(W)$, the ratio between the largest and the smallest singular value of $W$. This implies that orthogonal weight matrices, for example, are optimally conditioned with respect to the $L^2$ metric. Thus, a notable and well-known consequence of Proposition 1 is that imposing orthogonality constraints on $W$ improves the robustness of the network [13, 27, 38, 50].

While orthogonal constraints are widely studied in the literature, orthogonal matrices are not the only optimally conditioned ones. In fact, $\mathrm{cond}_2(W) = 1$ for any $W$ with constant singular values. In the next section, we will use this observation to design a low-rank and low-cost algorithm that trains well-conditioned networks by ensuring $\mathrm{cond}_2(W) \leq 1 + \tau$, for all layers $W$ and a desired tolerance $\tau > 0$.

## 4  Robust low-rank training

### 4.1  Instability of low-rank networks

Low-rank methods are popular strategies to reduce the memory storage and the computational cost of both training and inference phases of deep learning models [29, 55, 71]. Leveraging the intrinsic

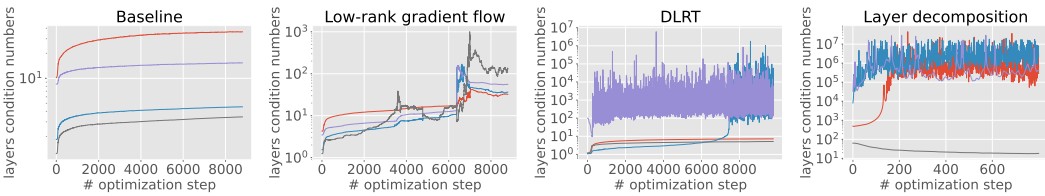

Figure 1: Evolution of layers' condition numbers during training for LeNet5 on MNIST. From left to right: standard full-rank baseline model; [71] vanilla low-rank training; [55] dynamical low-rank training based on gradient flow; [75] low-rank training through regularization. All low-rank training strategies are set to $80\%$ compression ratio (percentage of removed parameters with respect to the full baseline model).

low-rank structure of parameter matrices [7, 17, 49, 59], these methods train a subnetwork with weight matrices parametrized as $W = USV^\top$, for "tall and skinny" matrices $U, V$ with $r$ columns, and a small $r \times r$ matrix $S$. Training low-rank weight matrices has proven to effectively reduce training parameters while retaining performance comparable to those of the full model. However, while a variety of contributions have analyzed and refined low-rank methods to match the full model's accuracy, the robust accuracy of low-rank models has been partially overlooked in the literature.

Here we observe that reducing the rank of the layer may actually deteriorate the network's robustness. We argue that this phenomenon is imputable to the exploding condition number of the network. In Figure 1 we plot the evolution of the condition number $\mathrm{cond}_2$ for the four internal layers of LeNet5 during training using different low-rank training strategies and compare them with the full model. While the condition number of the full model grows moderately with the iteration count, the condition number of low-rank layers blows up drastically. This singular value instability leads to poor robustness performance of the methods, as observed in the experimental evaluation of Section 5.

In the following, we design a low-rank training model that allows imposing simple yet effective training constraints, bounding the condition number of the trained network to a desired tolerance $1 + \tau$, and improving the network robustness without affecting training nor inference costs.

### 4.2   Low-rank gradient flow with bounded singular values

Let $W \in \mathbb{R}^{n \times m}$ be the weight matrix of a linear layer within $f$. For an integer $r \leq \min\{m, n\}$ let $\mathcal{M}_r = \{W : \mathrm{rank}(W) = r\}$ be the manifold of rank-$r$ matrices which we parametrize as

$$\mathcal{M}_r = \left\{ USV^\top : U \in \mathbb{R}^{n \times r}, V \in \mathbb{R}^{m \times r} \text{ with orthonormal columns, } S \in \mathbb{R}^{r \times r} \text{ invertible} \right\}.$$

Obviously, the singular values of $W = USV^\top \in \mathcal{M}_r$ coincide the singular values of $S$. For $s, \varepsilon$ such that $0 < \varepsilon < s$, define $\Sigma_s(\varepsilon)$ as the set of matrices with singular values in the interval $[s - \varepsilon, s + \varepsilon]$. Note that $\Sigma_s(0)$ is a Riemannian manifold obtained essentially by an $s$ scaling of the standard Stiefel manifold (the manifold of matrices with orthonormal columns) and any $A \in \Sigma_s(0)$ is optimally conditioned, i.e. $\mathrm{cond}_2(A) = 1$. Thus, $\varepsilon$ can be interpreted as an approximation parameter that controls how close $\Sigma_s(\varepsilon)$ is to the "optimal" manifold $\Sigma_s(0)$. To enhance the network robustness, in the following we will constrain the parameter weight matrix $S$ to $\Sigma_s(\varepsilon)$. With this constraint, we get $\mathrm{cond}_2(W) \leq (s - \varepsilon)^{-1}(s + \varepsilon) = 1 + \tau$, with $\tau = 2(s - \varepsilon)^{-1}\varepsilon$, so that the tolerance $\tau$ on the network's conditioning can be tuned by suitably choosing the approximation parameter $\varepsilon$.

Given the loss function $\mathcal{L}$, we are interested in the constrained optimization problem

$$\min \mathcal{L} \qquad \text{s.t.} \qquad W = USV^\top \in \mathcal{M}_r \text{ and } S \in \Sigma_s(\varepsilon), \text{ for all layers } W . \tag{3}$$

To approach (3), we use standard arguments from geometric integration theory [31, 70] to design a training scheme that updates only the factors $U, S, V$ and the gradient of $\mathcal{L}$ with respect to $U, S, V$, without ever forming the full weights nor the full gradients. To this end, following [55], we first recast the optimization of $\mathcal{L}$ with respect to each layer $W$ as a continuous-time gradient flow

$$\dot{W}(t) = -\nabla_W \mathcal{L}(W(t)), \tag{4}$$

where "dot" denotes the time derivative and where we write $\mathcal{L}$ as a function of $W$ only, for brevity. Along the solution of the differential equation above, the loss decreases and a stationary point is

**Algorithm 1:** Pseudocode of robust well-Conditioned Low-Rank (**CondLR** ) training scheme

---

**Input:** Chosen compression rate, i.e. for each layer $W$ choose a rank $r$;
    Initial layers' weights parametrized as $W = USV^\top$, with $S \sim r \times r$;
    Second singular value moment of $S$, $s = \sqrt{\sum_k s_k(S)^2/r}$
    Conditioning tolerance $\tau > 0$

1 **for** each iteration and each layer **do** *(each block in parallel)*
2   $U \leftarrow$ one optimization step with gradient $G_1$ and initial point $U$
3   $U \leftarrow$ project $U$ on Stiefel manifold with $r$ orthonormal columns

4   $V \leftarrow$ one optimization step with gradient $G_2$ and initial point $V$
5   $V \leftarrow$ project $V$ on Stiefel manifold with $r$ orthonormal columns

6   $S \leftarrow$ one optimization step with gradient $G_3$ and initial point $S$
7   $s \leftarrow \sqrt{\sum_k s_k(S)^2/r}$, squareroot of second moment of the singular values of $S$
8   $\varepsilon \leftarrow \tau s/(2+\tau)$
9   $S \leftarrow$ project $S$ onto $\Sigma_s(\varepsilon)$

---

approached as $t \to \infty$. Now, if we assume $W \in \mathcal{M}_r$, then $\dot{W} \in T_W\mathcal{M}_r$, the tangent space of $\mathcal{M}_r$ at the point $W$. Thus, to ensure the whole trajectory $W(t) \in \mathcal{M}_r$, we can consider the projected gradient flow $\dot{W}(t) = -P_{W(t)}\nabla_W\mathcal{L}(W(t))$, where $P_W$ denotes the orthogonal projection (in the ambient space of matrices) onto $T_W\mathcal{M}_r$. Next, we notice that the projection $P_W\nabla_W\mathcal{L}$ can be defined by imposing orthogonality with respect to any point $Y \in T_W\mathcal{M}_r$, namely

$$\langle P_W\nabla_W\mathcal{L} - \nabla_W\mathcal{L}, Y \rangle = 0 \qquad \text{for all } Y \in T_W\mathcal{M}_r$$

where $\langle \cdot, \cdot \rangle$ is the Frobenius inner product. As discussed in e.g. [31, 55], the above equations combined with the well-known representation of $T_W\mathcal{M}_r$ yield a system of three gradient flow equations for the individual factors

$$\begin{cases} \dot{U} = -G_1(U), & G_1(U) = P_U^\perp\nabla_U\mathcal{L}(USV^\top)(SS^\top)^{-1} \\ \dot{V} = -G_2(V), & G_2(V) = P_V^\perp\nabla_V\mathcal{L}(USV^\top)(S^\top S)^{-\top} \\ \dot{S} = -G_3(S), & G_3(S) = \nabla_S\mathcal{L}(USV^\top) \end{cases} \tag{5}$$

where $P_U^\perp = (I - UU^\top)$ and $P_V^\perp = (I - VV^\top)$ are the projection operators onto the space orthogonal to the span of $U$ and $V$, respectively.

Based on the system of gradient flows above, we propose a training scheme that at each iteration and for each layer parametrized by the tuple $\{U, S, V\}$ proceeds as follows:

1. update $U$ and $V$ by numerically integrating the gradient flows $\dot{U} = -G_1(U)$ and $\dot{V} = -G_2(V)$
2. project the resulting $U, V$ onto the Stiefel manifold of matrices with $r$ orthonormal columns
3. update the $r \times r$ weight $S$ by integrating $\dot{S} = -G_3(S)$
4. for a fixed robustness tolerance $\tau$, project the computed $S$ onto $\Sigma_s(\varepsilon)$, choosing $s$ and $\varepsilon$ so that
   - $s$ is the best constant approximation to $S^\top S$, i.e. $s = \text{argmin}_\alpha \|S^\top S - \alpha^2 I\|_F$
   - $\varepsilon$ is such that the $\text{cond}_2$ of the projection of $S$ does not exceed $1 + \tau$

Note that the coefficients $s, \varepsilon$ at point 4 can be obtained explicitly by setting $s = \sqrt{\sum_{j=1}^r s_j(S)^2/r}$, the second moment of the singular values $s_j(S)$ of $S$, and $\varepsilon = \tau s/(2+\tau)$. Note also that, in the differential equations for $U, S, V$ in (5), the four steps above can be implemented in parallel for each of the three variables. The detailed pseudocode of the training scheme is presented in Algorithm 1. We conclude with several remarks about its implementation.

### Remarks, implementation details, and limitations

Each step of Algorithm 1 requires three optimization steps at lines 2, 4, 6. These steps can be implemented using standard first-order optimizers such as SGD with momentum or ADAM. Standard techniques can be used to project onto the Stiefel manifold at lines 3 and 5 of Algorithm 1, see e.g. [4, 70]. Here, we use the QR decomposition. As for the projection onto $\Sigma_s(\varepsilon)$ at line 9, we

compute the SVD of the small factor $S$ and set to $s + \varepsilon$ or $s - \varepsilon$ the singular values that fall outside the interval $[s - \varepsilon, s + \varepsilon]$. Note that, when $\tau = 0$, i.e. when we require perfect conditioning for the layer weight $W = USV^\top$, then the SVD of $S$ can be replaced by a QR step or any other Stiefel manifold projection. Indeed, we can equivalently set $s = \sqrt{\text{trace}(S^\top S)/r}$, and then project onto $\Sigma_s(0)$ by rescaling by a factor $s$ the projection of $S$ onto the Stiefel manifold. In this case, the system (5) further simplifies, as we can replace $(SS^\top)^{-1}$ and $(S^\top S)^{-\top}$ with the scalar $1/s^2$.

Overall, the compressed low-rank network has $r(n + m + r)$ parameters per each layer, where $n$ and $m$ are the number of input and output neurons. Thus, choosing $r$ so that $1 - r(n + m + r)/(nm) = \alpha$ can yield a desired compression rate $0 < \alpha < 1$ on the number of network parameters, i.e. the number of parameters one eliminates with respect to the full baseline. For example, in our experiments we will choose $r$ so that $\alpha = 0.5$ or $\alpha = 0.8$.

**Computational complexity.** Each pass of Alg.1 is done against a batch $x_{\text{batch}}$. In order to obtain minimal computational costs for each step in the algorithm, we evaluate $USV^\top x_{\text{batch}}$ sequentially: first $v = V^\top x_{\text{batch}}$, then $u = Sv$, and finally $Uu$. Assuming the size of the batch is negligible with respect to $n$ and $m$, the cost of these steps is $O(rm), O(r), O(rn)$, respectively. Adding the bias term and evaluating the activation function requires $O(n)$ operations. Hence, overall we have a cost per layer of $O(r(n + m + 1))$. Taping the forward evaluation to compute the gradient with respect to $U, S, V$ does not affect the asymptotic costs. The QR decompositions used for $U$ and $V$ require $O(r^2 n)$ and $O(r^2 m)$ operations respectively, $O(r^2(n + m))$ overall. Finally, computing the SVD in the projection step for $S$ requires a worst-case cost of $O(r^3)$. Hence the overall cost per layer is $O(r(1 + r)(n + m) + r^3)$ as opposed to the dense network training, which requires $O(nm)$ operations per layer. If $r \ll n, m$ then the low-rank method is cheaper than the full baseline. For example, if $n = m$, this happens provided $r < \sqrt{n}$.

**Limitations.** As the rank parameter $r$ has to be chosen a-priori for each layer of the network, a limitation of the proposed approach is the potential need for fine-tuning such parameter, even though the proposed analysis in Table 1 shows competitive performance for both $50\%$ and $80\%$ compression rates. Also, if the layer size $n \times m$ is not large enough, the compression ratio $\text{cr} = 1 - r(n + m + r)/(nm)$ might be limited. Thus the method works well only for wide-enough networks ($n, m \gg r$, so that $\text{cr} > 0$). Finally, a standard way to obtain better adversarial performance would be to combine the proposed conditioning-based robustness with adversarial training strategies [16, 63, 73]. However, the cost of producing adversarial examples during training is not negligible, especially when based on multi-step attacks, and thus the way to incorporate adversarial training without affecting the benefits obtained with low-rank compression is not straightforward.

## 4.3 Approximation guarantees

Optimization methods over the manifold of low-rank matrices are well-known to be affected by the stiff intrinsic geometry of the constraint manifold which has very high curvature around points where $W \in \mathcal{M}_r$ is almost singular [4, 31, 70]. This implies that even very small changes in $W$ may yield very different tangent spaces, and thus different training trajectories, as shown by the result below:

**Lemma 1** (Curvature bound, Lemma 4.2 [31]). *For $W \in \mathcal{M}_r$ let $s_{\min}(W) > 0$ be its smallest singular value. For any $W' \in \mathcal{M}_r$ arbitrarily close to $W$ and any matrix $B$, it holds*

$$\|P_W B - P_{W'} B\|_F \leq C \, s_{\min}(W)^{-1} \|W - W'\|_F \, ,$$

*where $C > 0$ depends only on $B$.*

In our gradient flow terminology, this phenomenon is shown by the presence of the matrix inversion in (5). While this is often an issue that may dramatically affect the performance of low-rank optimizers (see also Section 5), the proposed regularization step that enforces bounded singular values allows us to move along paths that avoid stiffness points. Using this observation, here we provide a bound on the quality of the low-rank neural network computed via Algorithm 1, provided there exists an optimal trajectory leading to an approximately low-rank network. We emphasize that this assumption is well-aligned with recent work showing the existence of high-performing low-rank nets in e.g. deep linear models [7, 17, 25, 49].

Assume the training is performed via gradient descent with learning rate $\lambda > 0$, and let $W(t)$ be the full gradient flow (4). Further, assume that for $t \in [0, \lambda]$ and a given $\varepsilon > 0$, for each layer there exists

$E(t)$ and $\widetilde{W}(t) \in \mathcal{M}_r \cap \Sigma_s(\varepsilon)$ such that

$$W(t) = \widetilde{W}(t) + E(t),$$

where $s$ is the second moment of the singular values of $W(t)$ and $E(t)$ is a perturbation that has bounded variation in time, namely $\|\dot{E}(t)\| \leq \eta$. In other words, we assume there exists a training trajectory that leads to an approximately low-rank weight matrix $W(t)$ with almost constant singular values. Because the value $s$ is bounded by construction, the parameter-dependent matrix $\widetilde{W}(t)$ possesses singular values exhibiting moderate lower bound. Thus, $W(t)$ is far from the stiffness region of (5) and we obtain the following bound, based on [31] (proof moved to Appendix C in the supplementary material)

**Theorem 1.** *Let $U_k S_k V_k^\top$ be a solution to* (5) *computed with $k$ steps of Algorithm 1. Assume that*
- *The low-rank initialization $U_0 S_0 V_0^\top$ coincides with the low-rank approximation $\widetilde{W}(0)$.*
- *The norm of the full gradient is bounded, i.e., $\|\nabla_W \mathcal{L}(W(t))\| \leq \mu$.*
- *The learning rate is bounded as $\lambda \leq \frac{s-\varepsilon}{4\sqrt{2}\mu\eta}$.*

*Then, assuming no numerical errors, the following error bound holds*

$$\|U_k S_k V_k^\top - W(\lambda k)\| \leq 3\lambda\eta.$$

Note that if $f$ is smooth enough (e.g. Lipschitz) then by the previous theorem we directly obtain an equivalent bound for the functional distance $\|f(U_k S_k V_k^\top) - f(W(\lambda k))\|$.

# 5  Experiments

We illustrate the performance of Algorithm 1 on a variety of test cases. All the experiments can be reproduced with the code in PyTorch available at `https://github.com/COMPiLELab/CondLR`. In order to assess the combined compression and robustness performance of the proposed method, we compare it against both full and low-rank baselines.

For all models, we compute natural accuracy and robust accuracy. Let $\{(x_i, y_i)\}_{i=1,\ldots,n}$ be the set of test images and the corresponding labels and let $f$ be the neural network model, with output $f(x)$ on the input $x$. We quantify the test set robust accuracy as:

$$\text{robust\_acc}(\delta) = \tfrac{1}{n} \sum_{i=1}^{n} \mathbb{1}_{\{y_i\}}(f(x_i + \delta_i))$$

where $\delta = (\delta_i)_{i=1,\ldots,n}$ are the adversarial perturbation associated to each sample. Notice that, in the unperturbed case with $\|\delta_i\| = 0$, the definition of robust accuracy exactly coincides with the definition of test accuracy. In our experiments, adversarial perturbations are produced by both the fast gradient sign method (FGSM) [19] and the projected gradient descent attack (PGD) [47], with $\|\delta_i\|_\infty = \epsilon$, and $\epsilon$ controls perturbation strength. As images in our set-up have input entries in $[0, 1]$, the perturbed input is then clamped to that interval. Note that, for the same reason, the value of $\epsilon$ controls in our case the relative size of the perturbation.

**Datasets.** We consider MNIST , CIFAR10, and CIFAR100 [33] datasets for evaluation purposes. The first contains 60,000 training images, the second one contains 50,000 training images, while the third one contains 50,000 training images. All the datasets have 10,000 test images, first two have 10 classes and last one has 100 classes. No data-augmentation is performed.

**Models.** We use LeNet5 [35] for MNIST dataset, VGG16 [58] and WideResnet (WRN16-4) [78] for CIFAR10 and CIFAR100. The general architecture for all the used networks is preserved across the models, while the weight-storing structures and optimization frameworks differ.

**Methods.** Our baseline network is the one done with the standard implementation. Cayley SGD [39] and Projected SGD [3] are Riemannian optimization-based methods that train the network weights over the Stiefel manifold of matrices with orthonormal columns. Thus, both methods ensure $\text{cond}_2(W) = 1$ for all layers. The former uses an iterative estimation of the Cayley transform, while the latter uses QR-based projection to retract the Riemannian gradient onto the Stiefel manifold. Both methods have no compression and use full-weight matrices. DLRT, SVD prune, and Vanilla are low-rank methods that ensure compression of the model parameters during training. DLRT [55] is based on a low-rank gradient flow model similar to the proposed Algorithm 1. SVD prune [75] is based on a regularized loss with a low-rank-promoting penalty term. This approach was designed to

compress the ranks of the network after training, but we imposed a fixed compression rate from the beginning for a fair comparison. "Vanilla" denotes the obvious low-rank approach, that parametrizes the layers as $W = UV^\top$ and performs alternate descent steps over $U$ and $V$, as done in e.g. [29, 71]. All models are implemented with fixed training compression ratios $\alpha = 0.5$ and $\alpha = 0.8$, i.e. we only use $50\%$ and $20\%$ of the parameters the full model would use during training, respectively. Finally, we implement CondLR as in Algorithm 1 for three choices of the conditioning tolerance $\tau \in \{0, 0.1, 0.5\}$. We also implement a modified version of Algorithm 1 in which $S$ is directly projected onto the Stiefel manifold $\Sigma_1(0)$, i.e. the parameter $s$ is fixed to one.

**Training.** Each method and model was trained for 120 epochs of stochastic gradient descent with a minibatch size of 128. We used a learning rate of 0.1 for LeNet5 and 0.05 for VGG16 with momentum 0.3 and 0.45, respectively, and a learning rate scheduler with factor = 0.3 at 70 and 100 epochs.

**Results.** For comparison, we measure robust accuracy for all chosen combinations of datasets, models, and methods using FGSM and PGD attacks with relative perturbation budget $\epsilon \in \{0.01, 0.02, 0.030.04, 0.05, 0.06\}$ for MNIST, $\epsilon \in \{0.001, 0.002, 0.003, 0.004, 0.005, 0.006\}$ for CIFAR10 VGG16, $\epsilon \in \{0.0003, 0.0006, 0.001, 0.0013, 0.0016\}$ for CIFAR10 and CIFAR100 with WRN16-4.

The results are summarized in Table 1 for LeNet5 MNIST and VGG16 CIFAR10 with FGSM and a subset of the perturbation budget, and in Table 2 for WRN16-4 CIFAR10 with both FGSM and PGD. The additional results are presented in Tables 3–6 and are moved to Appendix A. In the tables, we highlight (in gray) the best-performing method for each range of compression rates $\alpha \in \{0\%, 50\%, 80\%\}$, and the best method overall (in bold). The experiments show that the proposed method meaningfully conserves the accuracy as compared to the baseline for all datasets; moreover, the robustness is improved from the baseline for $50\%$ compression rate and, additionally, in the case of MNIST for $80\%$ compression rate. Compared to orthogonal non-compressed methods, CondLR with $50\%$ compression rate also outperforms Cayley SGD and Projected SGD. At the same time CondLR is compressed by design, so we reduce memory costs alongside gaining robustness. As anticipated in Section 4.1, low-rank methods without regularization deteriorate the condition number and thus the robustness of the model. This is consistent with the results of Table 1, where we observe that compression without regularization does not work well in terms of robust accuracy, in particular DLRT, Vanilla and SVD prune exhibit a drop in the performance in comparison with baseline and, consequently, CondLR . Specifically, per each fixed compression ratio $\alpha$, our method exhibits the highest robustness; moreover, the robustness of CondLR with $\alpha = 0.8$ is higher than the robustness of any other method with an even lower compression ratio, $\alpha = 0.5$.

As a further experimental evaluation, we analyze the robustness of low-rank training models with respect to small singular values. As shown by Theorem 1, CondLR is not affected by the steep curvature of the low-rank manifold $\mathcal{M}_r$, i.e. accuracy and convergence rate of CondLR do not deteriorate when the conditioning of the weight matrices explodes. In contrast, small singular values may significantly affect alternative low-rank training models. In Figure 2 we show the behavior of loss, accuracy, and condition number as functions of the iteration steps, when the network is initialized with ill-conditioned layer matrices. Precisely, similar to what is done in [29], we randomly sample Gaussian initial weights, compute their SVD to define the initial $U$ and $V$ factors, and then force the singular values of the initial $S$ factor to decay exponentially. We observe that CondLR and DLRT (which is also based on a low-rank gradient flow formulation) are the most robust, while the performance of the other methods deteriorates dramatically. This confirms that, as also observed in [29, 55], most low-rank approaches require specific fine-tuned choices of the initialization, while the proposed robust low-rank training model ensures solid performance independent of the initialization.

## 6 Conclusions

In recent years, extensive effort has been put into (a) studying different forms of implicit bias in deep learning, including bias towards low-rank parameter weights [7, 9, 18], and (b) designing modified training techniques that take advantage of the implicit low-rank structure to reduce training costs and compress the network [29, 55, 71, 75]. Based on the notion of condition number of a network, as a form of local Lipschitz constant, in this work we observe that training directly on the manifold of matrices with a fixed rank may lead to instabilities due to bad conditioning, going against the intuition that low-rank networks may be more robust to adversarial attacks. Thus, we propose an

algorithm able to mitigate this phenomenon in a computationally affordable way, allowing us to train directly on the manifold of matrices of a fixed rank while controlling the condition number.

Table 1: Method comparison results

| | | LeNet5 MNIST | | | | VGG16 Cifar10 | | | | c.r. (%) |
|---|---|---|---|---|---|---|---|---|---|---|
| Rel. perturbation $\epsilon$ | | 0.0 | 0.02 | 0.04 | 0.06 | 0.0 | 0.002 | 0.004 | 0.006 | |
| Baseline | | 0.9872 | 0.9793 | 0.9655 | 0.9407 | 0.9104 | **0.752** | 0.5822 | 0.4592 | 0 |
| Cayley SGD | | 0.9874 | 0.9804 | 0.9688 | 0.9486 | 0.8962 | 0.7446 | 0.5816 | 0.4529 | 0 |
| Projected SGD | | 0.9878 | 0.9807 | 0.968 | 0.9476 | 0.897 | 0.7455 | 0.5832 | 0.4574 | 0 |
| CondLR | $\tau = 0$ | **0.9883** | 0.9825 | 0.9732 | 0.958 | 0.9099 | 0.7456 | 0.5711 | 0.4292 | 50 |
| | $\tau = 0.1$ | 0.9877 | **0.9828** | **0.9763** | **0.9677** | 0.9093 | 0.7411 | 0.5985 | 0.4878 | 50 |
| | $\tau = 0.5$ | 0.9867 | 0.9802 | 0.9724 | 0.9598 | 0.8997 | 0.7225 | **0.6019** | **0.5017** | 50 |
| | stief | 0.986 | 0.9809 | 0.9721 | 0.9586 | **0.9138** | 0.7322 | 0.546 | 0.4054 | 50 |
| DLRT | | 0.967 | 0.9573 | 0.939 | 0.9078 | 0.8425 | 0.599 | 0.441 | 0.3691 | 50 |
| Vanilla | | 0.9875 | 0.9773 | 0.9603 | 0.94 | 0.8997 | 0.6771 | 0.4886 | 0.3849 | 50 |
| SVD prune | | 0.9883 | 0.9793 | 0.9639 | 0.9414 | 0.8992 | 0.673 | 0.4777 | 0.3698 | 50 |
| CondLR | $\tau = 0.0$ | 0.9882 | 0.9801 | 0.9676 | 0.9452 | 0.9066 | 0.7263 | 0.541 | 0.4 | 80 |
| | $\tau = 0.1$ | 0.9877 | 0.9795 | 0.966 | 0.9444 | 0.9048 | 0.7123 | 0.5262 | 0.4013 | 80 |
| | $\tau = 0.5$ | 0.9858 | 0.9768 | 0.9613 | 0.9342 | 0.8933 | 0.6823 | 0.4854 | 0.3666 | 80 |
| | stief | 0.9815 | 0.9729 | 0.9581 | 0.9399 | 0.9067 | 0.7184 | 0.5289 | 0.3861 | 80 |
| DLRT | | 0.9649 | 0.9517 | 0.9281 | 0.8865 | 0.8092 | 0.5839 | 0.4178 | 0.3086 | 80 |
| Vanilla | | 0.9862 | 0.972 | 0.9464 | 0.9194 | 0.881 | 0.6424 | 0.4266 | 0.299 | 80 |
| SVD prune | | 0.9864 | 0.9737 | 0.9512 | 0.9281 | 0.8799 | 0.6357 | 0.4206 | 0.2927 | 80 |

Table 2: WRN16-4 Cifar10

| Rel. perturbation $\epsilon$ | | 0.0 | 0.0003 | 0.0006 | 0.001 | 0.0013 | 0.0016 | cr (%) |
|---|---|---|---|---|---|---|---|---|
| Baseline, FGSM | | 0.9129 | 0.885 | 0.8531 | 0.8056 | 0.7667 | **0.7263** | 0 |
| CondLR | $\tau = 0.0$ | 0.9254 | 0.8958 | **0.865** | 0.8092 | 0.7669 | 0.7236 | 50 |
| | $\tau = 0.1$ | **0.9271** | **0.8974** | 0.8612 | **0.8117** | **0.7697** | 0.7223 | 50 |
| | $\tau = 0.0$ | 0.9106 | 0.8807 | 0.844 | 0.7874 | 0.7413 | 0.697 | 80 |
| | $\tau = 0.1$ | 0.9146 | 0.8833 | 0.8497 | 0.8004 | 0.7568 | 0.7142 | 80 |
| Baseline, PGD10 | | 0.9129 | 0.8898 | 0.8619 | 0.8188 | **0.7828** | **0.7435** | 0 |
| CondLR | $\tau = 0.0$ | 0.9254 | 0.8997 | **0.8714** | **0.8241** | 0.7791 | 0.7375 | 50 |
| | $\tau = 0.1$ | **0.9271** | **0.9018** | 0.8684 | 0.8233 | 0.782 | 0.7374 | 50 |
| | $\tau = 0.0$ | 0.9106 | 0.8838 | 0.8512 | 0.8014 | 0.7565 | 0.7064 | 80 |
| | $\tau = 0.1$ | 0.9146 | 0.8861 | 0.8582 | 0.813 | 0.7705 | 0.724 | 80 |

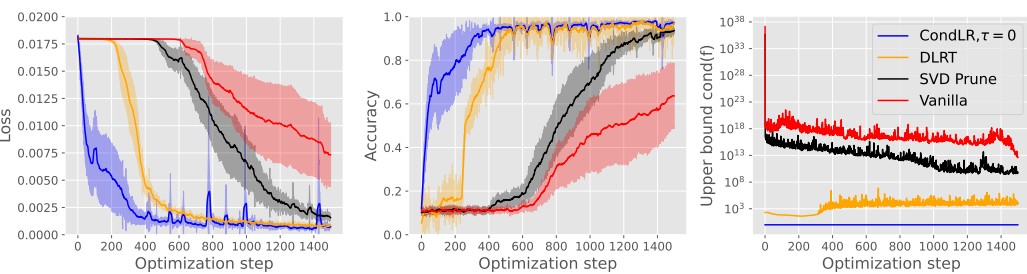

Figure 2: Evolution of loss, accuracy, and $\prod_i \text{cond}(W_i)$ for Lenet5 on MNIST dataset, for ill-conditioned initial layers whose singular values are forced to decay exponentially with powers of two.

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

# A  Additional results

In this section, we report the results on natural and robust accuracy for the broader range of perturbation budgets for Table 1 $\epsilon \in \{0.01, 0.02, 0.03, 0.04, 0.05, 0.06\}$ for LeNet5 on MNIST, and $\epsilon \in \{0.001, 0.002, 0.003, 0.004, 0.005, 0.006\}$ for VGG16 on CIFAR10. See Table 3 and Table 4 (standard deviation is shown in gray). Additionally we provide results for Projected Gradient Descent attack with L2 norm and 10 steps for VGG16 on CIFAR10 with perturbation budget $\epsilon \in \{0.1, 0.13, 0.16, 0.2, 0.23, 0.26, 0.3\}$. See Table 5. The results confirm the same findings as the ones reported in the main paper. Table 6 provides results of comparison between baseline and Algorithm 1 for WideResnet model with depth 16 and width 4 on CIFAR100 for both FGSM and PGD attack with $L\infty$ norm with perturbation budget $\epsilon \in \{0.0003, 0.0006, 0.001, 0.0013, 0.0016\}$.

# B  Proof of Proposition 1

**Lemma 2.** *Let* $\phi : (Y, \|\cdot\|_Y) \to (Z, \|\cdot\|_Z)$ *and* $\psi : (X, \|\cdot\|_X) \to (Y, \|\cdot\|_Y)$ *be continuous mappings between finite-dimensional Banach spaces, and assume* $\mathrm{cond}(\psi)$ *and* $\mathrm{cond}(\phi)$ *are well defined. Then the following inequality holds:*

$$\mathrm{cond}(\phi \circ \psi) \le \mathrm{cond}(\phi)\,\mathrm{cond}(\psi)$$

*Proof.* To prove this lemma, we will pass through the definition of $R(\phi \circ \psi, x, \delta)$. Let us recall that it is defined as

$$R(\phi \circ \psi, x, \delta) = \frac{\|\phi \circ \psi(x + \delta) - \phi \circ \psi(x)\|_Z \|x\|_X}{\|\delta\|_X \|\phi \circ \psi(x)\|_Z}$$

Now, if $\psi(x + \delta) - \psi(x) = 0$ for $\delta \ne 0$, then $R(\phi \circ \psi, x, \delta) = 0$. Since we're interested in the supremum, we can restrict to $\delta$ such that $\psi(x + \delta) - \psi(x) \ne 0$. Moreover, by multiplying and

Table 3: LeNet5 MNIST, FGSM

| | Rel. perturbation $\epsilon$ | 0.0 | 0.001 | 0.002 | 0.003 | 0.004 | 0.005 | 0.006 | cr (%) |
|---|---|---|---|---|---|---|---|---|---|
| | Baseline | 0.9872 
 8.1e-04 | 0.9836 
 5.e-04 | 0.9793 
 7.7e-04 | 0.9735 
 6.4e-04 | 0.9655 
 5.1e-04 | 0.9543 
 1.0e-03 | 0.9407 
 1.7e-03 | 0 |
| | Cayley SGD | 0.9874 
 3.8e-04 | 0.9843 
 5.2e-04 | 0.9804 
 7.1e-04 | 0.9753 
 4.9e-04 | 0.9688 
 5.1e-04 | 0.9602 
 7.4e-04 | 0.9486 
 1.3e-03 | 0 |
| | Projected SGD | 0.9878 
 2.1e-04 | 0.9845 
 5.2e-04 | 0.9807 
 8.6e-04 | 0.9753 
 7.8e-04 | 0.968 
 1.0e-03 | 0.9594 
 8.1e-04 | 0.9476 
 1.1e-03 | 0 |
| CondLR | $\tau = 0.0$ | **0.9883** 
 3.2e-04 | **0.9857** 
 8.5e-04 | 0.9825 
 4.9e-04 | 0.9783 
 9.9e-04 | 0.9732 
 9.e-04 | 0.9661 
 4.8e-04 | 0.958 
 6.0e-04 | 50 |
| | $\tau = 0.1$ | 0.9877 
 4.1e-04 | 0.9854 
 5.5e-04 | **0.9828** 
 8.4e-04 | **0.98** 
 1.2e-03 | **0.9763** 
 8.9e-04 | **0.972** 
 4.9e-04 | **0.9677** 
 1.2e-03 | 50 |
| | $\tau = 0.5$ | 0.9867 
 4.4e-04 | 0.9831 
 5.1e-04 | 0.9802 
 6.4e-04 | 0.9772 
 5.1e-04 | 0.9724 
 6.4e-04 | 0.9675 
 1.2e-03 | 0.9598 
 1.2e-03 | 50 |
| | stief | 0.986 
 2.6e-04 | 0.9838 
 2.4e-04 | 0.9809 
 2.4e-04 | 0.9773 
 2.8e-04 | 0.9721 
 3.6e-04 | 0.9655 
 6.2e-04 | 0.9586 
 1.4e-04 | 50 |
| | DLRT | 0.967 
 6.8e-04 | 0.9627 
 6.9e-04 | 0.9573 
 7.5e-04 | 0.9499 
 1.0e-03 | 0.939 
 6.6e-04 | 0.9253 
 1.1e-03 | 0.9078 
 1.5e-03 | 50 |
| | vanilla | 0.9875 
 5.8e-04 | 0.983 
 8.2e-04 | 0.9773 
 1.2e-03 | 0.97 
 4.1e-03 | 0.9603 
 4.1e-03 | 0.9503 
 4.2e-03 | 0.94 
 3.9e-03 | 50 |
| | SVD prune | 0.9883 
 9.6e-04 | 0.9841 
 8.6e-04 | 0.9793 
 1.1e-03 | 0.9733 
 1.4e-03 | 0.9639 
 1.2e-03 | 0.9526 
 1.5e-03 | 0.9414 
 2.6e-03 | 50 |
| CondLR | $\tau = 0.0$ | 0.9882 
 7.5e-04 | 0.9846 
 2.0e-03 | 0.9801 
 3.4e-03 | 0.9743 
 4.3e-03 | 0.9676 
 6.3e-03 | 0.958 
 8.e-03 | 0.9452 
 1.2e-02 | 80 |
| | $\tau = 0.1$ | 0.9877 
 5.1e-04 | 0.984 
 5.3e-04 | 0.9795 
 6.1e-04 | 0.9736 
 1.0e-03 | 0.966 
 8.1e-04 | 0.9564 
 1.4e-03 | 0.9444 
 1.8e-03 | 80 |
| | $\tau = 0.5$ | 0.9858 
 6.5e-04 | 0.982 
 8.e-04 | 0.9768 
 8.2e-04 | 0.97 
 1.3e-03 | 0.9613 
 1.9e-03 | 0.9487 
 3.5e-03 | 0.9342 
 4.0e-03 | 80 |
| | stief | 0.9815 
 3.8e-04 | 0.9779 
 4.6e-04 | 0.9729 
 1.2e-04 | 0.9659 
 3.7e-04 | 0.9581 
 5.7e-04 | 0.9499 
 4.5e-04 | 0.9399 
 1.0e-03 | 80 |
| | DLRT | 0.9649 
 5.8e-04 | 0.9596 
 9.7e-04 | 0.9517 
 1.7e-03 | 0.9414 
 2.6e-03 | 0.9281 
 4.6e-03 | 0.9108 
 7.6e-03 | 0.8865 
 1.1e-02 | 80 |
| | vanilla | 0.9862 
 6.9e-04 | 0.9802 
 9.7e-04 | 0.972 
 1.9e-03 | 0.9602 
 3.4e-03 | 0.9464 
 3.1e-03 | 0.9331 
 4.1e-03 | 0.9194 
 5.2e-03 | 80 |
| | SVD prune | 0.9864 
 8.0e-04 | 0.981 
 1.1e-03 | 0.9737 
 2.2e-03 | 0.9634 
 3.e-03 | 0.9512 
 4.5e-03 | 0.9392 
 5.1e-03 | 0.9281 
 5.9e-03 | 80 |

dividing by $\|\psi(x)\|_Y$ we obtain:

$$R(\phi \circ \psi, x, \delta) = \left( \frac{\|\phi \circ \psi(x+\delta) - \phi \circ \psi(x)\|_Z \|\psi(x)\|_Y}{\|\psi(x+\delta) - \psi(x)\|_Y \|\phi \circ \psi(x)\|_Z} \right) \left( \frac{\|\psi(x+\delta) - \psi(x)\|_Y \|x\|_X}{\|\delta\|_X \|\psi(x)\|_Y} \right)$$

Now, if we define $\mathrm{cond}(f, x, \varepsilon) = \sup\limits_{\delta \neq 0 : \|\delta\|_X \leq \varepsilon} R(f, x, \delta)$, we take the supremum both sides on the set

$\{\delta \in X \mid \|\delta\|_X \leq \varepsilon\}$ and we can upper bound it with the product of the suprema of the two blocks

$$\sup_{\delta \neq 0 : \|\delta\|_X \leq \varepsilon} R(\phi \circ \psi, x, \delta) = \mathrm{cond}(\psi, x, \varepsilon) \sup_{\delta \neq 0 : \|\delta\|_X \leq \varepsilon} \left( \frac{\|\phi \circ \psi(x+\delta) - \phi \circ \psi(x)\|_Z \|\psi(x)\|_Y}{\|\psi(x+\delta) - \psi(x)\|_Y \|\phi \circ \psi(x)\|_Z} \right) = \star$$

Now, letting $\eta = \psi(x+\delta) - \psi(\delta)$, we can rewrite the second part of the last equation as a sort of restriction of $\mathrm{cond}(\phi, \psi(x), \varepsilon)$ to the particular set of perturbation directions of the form $\eta$. Thus, we can lower-bound it as:

$$\star \leq \mathrm{cond}(\psi, x, \varepsilon) \sup_{\eta \neq 0 : \|\eta\|_Y \leq \varepsilon} \left( \frac{\|\phi(\psi(x) + \eta) - \phi(\psi(x))\|_Z \|\psi(x)\|_Y}{\|\eta\|_Y \|\phi(\psi(x))\|_Z} \right) =$$

$$= \mathrm{cond}(\psi, x, \varepsilon) \, \mathrm{cond}(\phi, \psi(x), \varepsilon)$$

By hypothesis, $\mathrm{cond}(\phi)$ and $\mathrm{cond}(\psi)$ are finite, so we can take the limit $\varepsilon \downarrow 0$ to get:

$$\mathrm{cond}(\phi \circ \psi, x) \leq \mathrm{cond}(\phi, \psi(x)) \, \mathrm{cond}(\psi, x)$$

Finally, taking the supremum over $x$ on both members of the last equation, we can upper bound it with the original condition number without constraint on the directions:

$$\mathrm{cond}(\phi \circ \psi) \leq \sup_x \mathrm{cond}(\phi, \psi(x)) \, \mathrm{cond}(\psi, x) \leq \mathrm{cond}(\phi) \, \mathrm{cond}(\psi)$$

and thus conclude. $\qquad \square$

Table 4: VGG16 Cifar10, FGSM

| | Rel. perturbation $\epsilon$ | 0.0 | 0.001 | 0.002 | 0.003 | 0.004 | 0.005 | 0.006 | cr (%) |
|---|---|---|---|---|---|---|---|---|---|
| | Baseline | 0.9104 1.3e-03 | **0.8397** 2.e-03 | **0.752** 3.3e-03 | 0.6631 2.7e-03 | 0.5822 3.9e-03 | 0.5154 6.6e-03 | 0.4592 8.9e-03 | 0 |
| | Cayley SGD | 0.8962 1.7e-03 | 0.8262 3.9e-03 | 0.7446 3.5e-03 | 0.6624 4.1e-03 | 0.5816 4.4e-03 | 0.5132 4.2e-03 | 0.4529 4.3e-03 | 0 |
| | Projected SGD | 0.897 1.8e-03 | 0.8271 3.1e-03 | 0.7455 3.0e-03 | 0.661 4.1e-03 | 0.5832 5.3e-03 | 0.5159 4.2e-03 | 0.4574 4.5e-03 | 0 |
| CondLR | $\tau = 0.0$ | 0.9099 7.4e-03 | 0.8366 9.2e-03 | 0.7456 9.8e-03 | 0.6564 1.4e-02 | 0.5711 1.5e-02 | 0.4955 1.8e-02 | 0.4292 1.6e-02 | 50 |
| CondLR | $\tau = 0.1$ | 0.9093 1.9e-03 | 0.8291 3.1e-03 | 0.7411 6.e-03 | **0.6678** 1.2e-02 | 0.5985 1.2e-02 | 0.5386 8.4e-03 | 0.4878 5.6e-03 | 50 |
| CondLR | $\tau = 0.5$ | 0.8997 1.0e-03 | 0.8136 1.9e-03 | 0.7225 5.3e-03 | 0.6577 7.5e-03 | **0.6019** 6.8e-03 | **0.5485** 7.7e-03 | **0.5017** 9.1e-03 | 50 |
| CondLR | stief | **0.9138** 1.3e-03 | 0.8331 3.3e-03 | 0.7322 5.4e-03 | 0.6337 3.9e-03 | 0.546 3.7e-03 | 0.4702 4.1e-03 | 0.4054 3.9e-03 | 50 |
| | DLRT | 0.8425 1.6e-03 | 0.7179 6.2e-03 | 0.599 8.8e-03 | 0.5065 8.3e-03 | 0.441 1.2e-02 | 0.3979 1.6e-02 | 0.3691 1.8e-02 | 50 |
| | vanilla | 0.8997 3.2e-03 | 0.7962 4.4e-03 | 0.6771 6.1e-03 | 0.5708 8.3e-03 | 0.4886 9.9e-03 | 0.4276 1.0e-02 | 0.3849 1.1e-02 | 50 |
| | SVD prune | 0.8992 3.1e-03 | 0.7936 1.7e-03 | 0.673 2.5e-03 | 0.5643 7.6e-03 | 0.4777 1.1e-02 | 0.4161 1.3e-02 | 0.3698 1.3e-02 | 50 |
| CondLR | $\tau = 0.0$ | 0.9066 2.2e-03 | 0.825 1.8e-03 | 0.7263 2.8e-03 | 0.6289 3.2e-03 | 0.541 5.2e-03 | 0.4633 5.3e-03 | 0.4 5.6e-03 | 80 |
| CondLR | $\tau = 0.1$ | 0.9048 1.6e-03 | 0.8152 3.8e-03 | 0.7123 5.4e-03 | 0.6112 6.3e-03 | 0.5262 9.1e-03 | 0.4574 9.9e-03 | 0.4013 8.6e-03 | 80 |
| CondLR | $\tau = 0.5$ | 0.8933 1.3e-03 | 0.7952 1.4e-03 | 0.6823 4.2e-03 | 0.5748 6.e-03 | 0.4854 5.3e-03 | 0.4177 5.3e-03 | 0.3666 5.3e-03 | 80 |
| CondLR | stief | 0.9067 4.4e-04 | 0.8202 2.5e-03 | 0.7184 5.9e-03 | 0.6185 5.5e-03 | 0.5289 7.6e-03 | 0.4515 8.9e-03 | 0.3861 9.3e-03 | 80 |
| | DLRT | 0.8092 2.e-02 | 0.6916 1.9e-02 | 0.5839 2.2e-02 | 0.4936 2.5e-02 | 0.4178 2.4e-02 | 0.3577 2.3e-02 | 0.3086 2.2e-02 | 80 |
| | vanilla | 0.881 2.4e-03 | 0.7687 5.e-03 | 0.6424 7.4e-03 | 0.523 8.5e-03 | 0.4266 1.1e-02 | 0.3533 1.2e-02 | 0.299 1.2e-02 | 80 |
| | SVD prune | 0.8799 3.7e-03 | 0.7634 8.2e-03 | 0.6357 1.2e-02 | 0.5168 1.4e-02 | 0.4206 1.2e-02 | 0.3474 1.1e-02 | 0.2927 1.1e-02 | 80 |

Now, the proof of Proposition 1 follows directly by applying the Lemma above recursively.

*Proof of Proposition1.* By defining $T_i(z) = W_i z$, we can write the neural network $f$ as

$$f(x) = (\sigma_L \circ T_L \circ \sigma_{L-1} \circ \cdots \circ T_1)(x)$$

for nonlinear activation functions $\sigma_i$. Thus, for $L = 1$, the thesis follows directly from Lemma2, as long as $\operatorname{cond}(\sigma_1) \leq C < \infty$. For $L > 1$, one can unwrap the compositional structure of $f$ from the left, defining $\phi(z) = (\sigma_L \circ T_L)(z)$ and $\psi(x) = (\sigma_{L-1} \circ \cdots \circ T_1)(x)$. Then by using Lemma2 we have that

$$\operatorname{cond}(f) \leq \operatorname{cond}(\phi) \operatorname{cond}(\psi) \leq \operatorname{cond}(\sigma_L) \operatorname{cond}(T_L) \operatorname{cond}(\psi).$$

Now, since $\psi$ is a network of depth $L - 1$, we can proceed inductively to obtain that $\operatorname{cond}(f) \leq C \prod_i \operatorname{cond}(T_i)$, with $C = \Pi_{i=1}^{L} \operatorname{cond}(\sigma_i)$, and we conclude.

$\square$

Note that the result above is of interest only if $\operatorname{cond}(\sigma) = \sup_{x \in \mathcal{X}} \operatorname{cond}(\sigma; x) < \infty$. When $\sigma$ is Lipschitz, using the formula

$$\operatorname{cond}(f; x) = \sup_{\nu_x \in \partial \sigma(x)} \|\nu_x\| \|x\| \|\sigma(x)\|^{-1},$$

Table 5: VGG16 Cifar10, PGD

| Rel. perturbation $\epsilon$ | | 0.0 | 0.1 | 0.13 | 0.16 | 0.2 | 0.23 | 0.26 | 0.3 | cr (%) |
|---|---|---|---|---|---|---|---|---|---|---|
| Baseline | | 0.9104 (1.3e-03) | 0.6704 (5.4e-03) | 0.5799 (8.1e-03) | 0.5021 (9.0e-03) | 0.4222 (1.4e-02) | 0.3788 (1.4e-02) | 0.347 (1.4e-02) | 0.3188 (1.4e-02) | 0 |
| Cayley SGD | | 0.8962 (1.7e-03) | 0.6768 (4.4e-03) | 0.5938 (5.3e-03) | 0.5109 (7.1e-03) | 0.4087 (6.9e-03) | 0.3446 (8.4e-03) | 0.2921 (8.9e-03) | 0.2362 (1.0e-02) | 0 |
| Projected SGD | | 0.897 (1.8e-03) | 0.6764 (5.0e-03) | 0.5925 (6.4e-03) | 0.5106 (5.6e-03) | 0.4086 (6.8e-03) | 0.3451 (9.e-03) | 0.2919 (7.5e-03) | 0.2364 (7.6e-03) | 0 |
| CondLR | $\tau = 0.0$ | 0.9099 (7.4e-03) | 0.6608 (1.8e-02) | 0.562 (2.4e-02) | 0.4712 (2.8e-02) | 0.3665 (2.7e-02) | 0.305 (2.2e-02) | 0.2551 (1.8e-02) | 0.2078 (1.5e-02) | 50 |
| | $\tau = 0.1$ | 0.9093 (1.9e-03) | 0.6703 (1.6e-02) | 0.6208 (3.4e-02) | 0.5915 (5.1e-02) | 0.5692 (6.5e-02) | 0.5596 (7.1e-02) | 0.5528 (7.6e-02) | 0.5458 (7.9e-02) | 50 |
| | $\tau = 0.5$ | 0.8997 (1.0e-03) | 0.6484 (9.8e-03) | 0.6025 (1.9e-02) | 0.5775 (2.6e-02) | 0.5603 (3.1e-02) | 0.5521 (3.2e-02) | 0.5475 (3.3e-02) | 0.5425 (3.5e-02) | 50 |
| | stief | 0.9138 (1.3e-03) | 0.6318 (6.1e-03) | 0.5253 (7.0e-03) | 0.4266 (6.9e-03) | 0.3118 (7.8e-03) | 0.2445 (6.4e-03) | 0.192 (6.9e-03) | 0.1388 (7.2e-03) | 50 |
| vanilla | | 0.8997 (3.2e-03) | 0.5574 (1.2e-02) | 0.4511 (1.5e-02) | 0.3686 (1.5e-02) | 0.2962 (1.8e-02) | 0.2628 (2.0e-02) | 0.2402 (2.1e-02) | 0.2184 (2.2e-02) | 50 |
| SVD prune | | 0.8992 (3.1e-03) | 0.5487 (5.4e-03) | 0.4385 (8.9e-03) | 0.3523 (1.5e-02) | 0.2795 (2.0e-02) | 0.2438 (2.5e-02) | 0.2206 (2.9e-02) | 0.1994 (3.2e-02) | 50 |
| CondLR | $\tau = 0.0$ | 0.9066 (2.2e-03) | 0.6324 (4.8e-03) | 0.5323 (6.2e-03) | 0.4357 (5.3e-03) | 0.325 (8.9e-03) | 0.2599 (1.2e-02) | 0.2096 (1.5e-02) | 0.1603 (1.5e-02) | 80 |
| | $\tau = 0.1$ | 0.9048 (1.6e-03) | 0.61 (8.7e-03) | 0.5084 (9.7e-03) | 0.4219 (1.1e-02) | 0.337 (1.1e-02) | 0.2944 (1.1e-02) | 0.2655 (1.1e-02) | 0.2397 (1.1e-02) | 80 |
| | $\tau = 0.5$ | 0.8933 (1.3e-03) | 0.5745 (9.1e-03) | 0.4635 (9.7e-03) | 0.372 (1.0e-02) | 0.283 (1.3e-02) | 0.2382 (1.3e-02) | 0.2065 (1.6e-02) | 0.1784 (1.7e-02) | 80 |
| | stief | 0.9067 (4.4e-04) | 0.6153 (6.7e-03) | 0.5094 (9.5e-03) | 0.4082 (1.e-02) | 0.2924 (9.4e-03) | 0.2236 (9.1e-03) | 0.1716 (8.7e-03) | 0.118 (7.5e-03) | 80 |
| vanilla | | 0.881 (2.4e-03) | 0.514 (1.3e-02) | 0.397 (1.5e-02) | 0.3012 (1.4e-02) | 0.2097 (1.3e-02) | 0.1629 (1.4e-02) | 0.1315 (1.4e-02) | 0.1037 (1.4e-02) | 80 |
| SVD prune | | 0.8799 (3.7e-03) | 0.5064 (2.e-02) | 0.3906 (2.1e-02) | 0.2957 (1.8e-02) | 0.2016 (1.5e-02) | 0.1549 (1.2e-02) | 0.1222 (1.1e-02) | 0.0957 (1.2e-02) | 80 |

Table 6: WRN16-4 Cifar100

| Rel. perturbation $\epsilon$ | | 0.0003 | 0.0006 | 0.001 | 0.0013 | 0.0016 | cr (%) |
|---|---|---|---|---|---|---|---|
| Baseline, FGSM | | 0.5845 | 0.5267 | 0.4606 | 0.4187 | 0.3841 | 0 |
| CondLR | $\tau = 0.1$ | 0.6492 | 0.5806 | 0.5009 | 0.4499 | 0.4011 | 50 |
| | $\tau = 0.5$ | 0.6062 | 0.5442 | 0.4693 | 0.4199 | 0.3774 | 50 |
| Baseline, PGD10 | | 0.5925 | 0.5401 | 0.4713 | 0.4289 | 0.3878 | 0 |
| CondLR | $\tau = 0.1$ | 0.6552 | 0.5918 | 0.5127 | 0.4531 | 0.3972] | 50 |
| | $\tau = 0.5$ | 0.6160 | 0.5547 | 0.4823 | 0.4290 | 0.3810 | 50 |

with $\partial$ being the Clarke's generalized gradient operator [14], we observe below that this is the case for a broad list of activation functions $\sigma$ and feature spaces $\mathcal{X}$.

- **LeakyReLU.** For $x \in \mathbb{R}, \alpha > 0$, let $\sigma(x) = \max\{0, x\} + \alpha \min\{0, x\}$. Then any $\nu_x \in \partial\sigma(x)$ is such that $\nu_x = 1$ if $x > 0$; $\nu_x = \alpha$ if $x < 0$; $\nu_x = [\min(\alpha, 1), \max(\alpha, 1)]$ otherwise. Thus

$$\text{cond}(\sigma) = \sup_{x \neq 0} \text{cond}(\sigma; x) = \sup_{x \neq 0} \sup_{\beta \in \partial\sigma(x)} \frac{|x||\mathbf{1}_{x>0} + \alpha\mathbf{1}_{x<0} + \beta\mathbf{1}_{x=0}|}{|\max\{0, x\} + \alpha \min\{0, x\}|} = \max(\alpha, 1)$$

- **Tanh.** For $x \in \mathbb{R}$, let $\sigma(x) = \tanh(x)$. Then $\sigma'(x) = \frac{1}{\cosh^2(x)}$ and thus

$$\text{cond}(\sigma) = \sup_x \frac{|x|}{|\tanh(x)||\cosh^2(x)|} = \sup_x \frac{|4x|}{|e^x - e^{-x}||e^x + e^{-x}|} = 1$$

Since the maximum of $\text{cond}(\sigma, x)$ is reached at zero, where the function can be extended by continuity.

- **Hardtanh.** For $x \in [-a, a]$ and $a > 0$, let $\sigma(x) = a\mathbf{1}_{x>a} - a\mathbf{1}_{x<-a} + x\mathbf{1}_{x \in [-a,a]}$. Then, we have that $\partial\sigma(x)$ coincides with the derivative values in all points but $x = \pm a$. In those two points, we have $\partial\sigma(\pm a) = [0, 1]$. Thus, for any $\nu_x \in \partial\sigma(x)$, we have

$$\text{cond}(\sigma) = \sup_{x \in [-a,a]} \frac{|\nu_x||x|}{|\sigma(x)|} \leq \sup_{x \in [-a,a]} \frac{|x|}{|\sigma(x)|} = a$$

- **Logistic sigmoid.** For $x \in \mathbb{R}$ let $\sigma(x) = (1 + e^{-x})^{-1}$. Then $\sigma'(x) = \sigma(x)(1 - \sigma(x))$ and thus

$$\text{cond}(\sigma; x) = |x|(1 - \sigma(x)) = |x|e^{-x}(1 + e^{-x})^{-1}.$$

Therefore, when $x \geq 0$, we have $|x|e^{-x} \leq 1/e$ and $(1 + e^{-x}) \geq 1$, thus $\text{cond}(\sigma; x) \leq 1/e$.

- **Softplus.** For $x \in \mathbb{R}$, let $\sigma(x) = \ln(1 + e^x)$. Then $\sigma'(x) = S(x) = (1 + e^{-x})^{-1}$ and $\text{cond}(\sigma; x) = |x|S(x)\sigma(x)^{-1}$. Thus, for $x \geq 0$, we have $\text{cond}(\sigma; x) \leq 1$.

- **SiLU.** For $x \in \mathbb{R}$ let $\sigma(x) = x(1 + e^{-x})^{-1} = xS(x)$. Then, $\sigma'(x) = S(x) + xS(x)(1 - S(x))$ and thus for any $x \geq 0$ we have

$$\text{cond}(\sigma; x) = |1 + x(1 - S(x))| \leq 1 + \frac{1}{e}$$

## C  Proof of Theorem 1

In the following, the proof of the main approximation result is presented. We underline that the core part of the proof relies on [31, Theorem 5.2]. For completeness, we repropose here main elements of the argument. We refer the interested reader to [31] and references therein for further details.

*Proof.* Let $Y(t)$ be the solution of (5) at time $t \in [0, \lambda]$. First, we observe that the projected subflows of $W(t) = \widetilde{W}(t) + E(t)$ and $\widetilde{W}(t)$ satisfy the differential equations

$$\begin{cases} \dot{Y} = P(Y)\dot{\widetilde{W}} + P(Y)\dot{E}, \\ \dot{\widetilde{W}} = P\left(\widetilde{W}\right)\dot{W}. \end{cases}$$

where $P(\cdot)$ denotes the orthogonal projection into the tangent space of the low-rank manifold $\mathcal{M}_r$. Next, we observe that the following identities hold

$$(P(Y) - P(\widetilde{W}))\dot{\widetilde{W}} = -(P^\perp(Y) - P^\perp(\widetilde{W}))\dot{\widetilde{W}} = -P^\perp(Y)\dot{\widetilde{W}} = -P^\perp(Y)^2\dot{\widetilde{W}}.$$

where $P^\perp(\cdot) = I - P(\cdot)$ represents the complementary orthogonal projection. The latter implies that

$$\langle Y - \widetilde{W}, (P(Y) - P(\widetilde{W}))\dot{\widetilde{W}} \rangle = \langle P^\perp(Y)(Y - \widetilde{W}), (P(Y) - P(\widetilde{W}))\dot{\widetilde{W}} \rangle.$$

Let $\gamma = 32\mu(s - \varepsilon)^{-2}$. It follows from [31, Lemma 4.2] that

$$\langle Y - \widetilde{W}, \dot{Y} - \dot{\widetilde{W}} \rangle = \langle P^\perp(Y)(Y - \widetilde{W}), (P(Y) - P(\widetilde{W}))\dot{\widetilde{W}} \rangle + \langle Y - \widetilde{W}, P(Y)\dot{E} \rangle$$

$$\leq \gamma\|Y - \widetilde{W}\|^3 + \eta\|Y - \widetilde{W}\|.$$

Further, we remind that

$$\langle Y - \widetilde{W}, \dot{Y} - \dot{\widetilde{W}} \rangle = \frac{1}{2}\frac{d}{dt}\|Y - \widetilde{W}\|^2 = \|Y - \widetilde{W}\|\frac{d}{dt}\|Y - \widetilde{W}\|.$$

Hence, the error $e(t) = \|Y(t) - \widetilde{W}(t)\|$ satisfies the differential inequality

$$\dot{e} \le \gamma e^2 + \eta, \quad e(0) = 0.$$

The error $e(t)$ for $t \in [0, \lambda]$ admits an upper bound given by the solution of

$$\dot{z} = \gamma z^2 + \eta, \quad z(0) = 0.$$

The last differential initial-value problem admits a closed solution given by

$$z(t) = \sqrt{\eta/\gamma}\tan\left(t\sqrt{\eta\gamma}\right),$$

where the last term is bounded by $2t\eta$ for $t\sqrt{\gamma\eta} \le 1$. The proof thus concludes as follows

$$\|Y(t) - W(t)\| \le \|Y(t) - \widetilde{W}(t)\| + \|E(t)\| \le 2t\eta + t\eta = 3t\eta,$$

where the last estimate arise by the integral identity $E(t) = \int_0^t \dot{E}(s)ds$. $\qquad\square$

