# OpenReview forum: "Robust low-rank training via approximate orthonormal constraints"
_NeurIPS.cc/2023/Conference — NeurIPS 2023 poster_

### Official Review · Reviewer_UZkq · 2023-06-10

**Soundness:** 4 excellent
**Presentation:** 4 excellent
**Contribution:** 3 good
**Rating:** 7
**Confidence:** 4

**Summary:**

This work demonstrates that conventional low-rank training methods suffer from poor adversarial robustness, which is hypothesized to stem from ill conditioning of weight matrices during training. The authors address this by constraining low-rank weight matrices to be approximately orthogonal.

**Strengths:**

I found this paper to be a solid contribution, although my lack of familiarity with low-rank approaches means I'm not exactly sure which parts of the paper are novel.

1. The observation in Figure 1 that low-rank training methods lead to exploding conditioning numbers is quite interesting and is not something I have seen before in the literature.
2. The empirical gains in the experimental section are substantial, even if only using a FGSM atatck.
3. The paper is very well-written and organized, with clear explanations and visualizations.
4. The approximation guarantee is a nice inclusion of theory to explain the numerical advantages of avoiding high-curvature stiffness points.

**Weaknesses:**

5. The observation behind S1 was left unexplained: why do low-rank constrained matrices lead to large condition numbers? This might be out of scope but felt somewhat unsatisfying.
6. The empirical evaluations only used the fast gradient sign method, which is a weak attack. This is my biggest reason for not giving an full accept rating -- I'd like to at least see PGD on this problem, although the results for FGSM are reasonably convincing.

**Questions:**

7. Precisely which parts of the algorithm in 4.2 are original? My impression is that the main contribution is step (4), the projection onto $\Sigma_s(\epsilon)$.
8. In proposition 1, X is referred to as the "feature space" -- is this a deliberate deviation from the more standard terminology of "input space"? Or am I missing something here.
9. Related to question 2, I'm a little confused about the notation in Proposition 1. The $cond(f; x)$ definition boils down to the function $f$ taking $x$ as an input; proposition 1 uses $cond(\sigma_i; x)$, even though $\sigma_i$ is at an intermediate layer in the network and does not directly take $x$ as an argument. My interpretation would be that this should read $\cond(\sigma_i, W_i z_i)$. It doesn't really affect anything downstream as we're just taking the sup over the argument to the condition number anyways, but the notation seems a bit imprecise to me as on the LHS of the inequality we're taking the sup over caligraphic X representing the input space, and on the right hand side it's the sup over caligraphic X representing the feature space, I feel that each intermediate feature space should have its own notation $X_i$ or something of the sort.
10. In 157 the condition number for logistic sigmoid requires nonnegative features, which does not seem to be satisfied in general for intermediate neural network layers.
11. In theorem 1, what is the significance of the phrase "assuming no numerical errors" -- is this referring to rounding errors, or something more serious like non-invertibility of a matrix during execution?
12. The definition of relative error in equation (1) doesn't sit perfectly right with me, as it is dependent on the normalization of the inputs. If all inputs were shifted by some large amount, the relative error would increase although nothing important about the data has changed. Also, the relative error at x=0 is always zero (or undefined?) which doesn't make a lot of sense to me. Would appreciate any clarity here.

**Limitations:**

Limitations are explicitly addressed in section 4.2.

---

> ### Author Rebuttal · Authors · 2023-08-09
>
> Thank you for the encouraging feedback. We have now implemented new experimental evaluation with PGD attack and on one additional network (wideresnet) on both cifar10 and cifar100. All new results are available in the pdf attached to the 'general rebuttal comment'.
>
> Re the specific questions:
>
> > The observation behind S1 was left unexplained: why do low-rank constrained matrices lead to large condition numbers? This might be out of scope but felt somewhat unsatisfying.
>
> This is a good point. At the moment we don't have additional theoretical nor experimental insight on this matter other than our guess (stated in §1):  the singular values grow very large in order to match the baseline accuracy and to compensate for the lack of parameters, while some small sing values are also still there as the network is compressed in low-rank format. We will investigate this question further in the future as we agree it is an important question that needs to be further explored.
>
> > Precisely which parts of the algorithm in 4.2 are original? My impression is that the main contribution is step (4), the projection onto $\Sigma_{s}(\epsilon)$.
>
> The equations that describe the gradient flow on the low-rank manifold (eq. (5) in the paper) are known in the geometric integration literature, but not broadly discussed in the ML community to our knowledge. Equations (5) often have instability issues, causing practitioners to avoid using them directly. The novelty of our approach lies in establishing a framework where the utilization of the provided formulation (5) is problem-free. This is thanks to the coupling with the projection onto a Stiefel-like manifold which has a dynamically changing barycenter. The projection (and the corresponding analysis of stability) and the structure of the chosen manifold are entirely novel.
>
> > In proposition 1, $X$ is referred to as the "feature space" --- is this a deliberate deviation from the more standard terminology of ``input space"? Or am I missing something here.
>
> Apologies, this is simply a typo. We will change this into 'input space'.
>
> > Related to question 2, I'm a little confused about the notation in Proposition 1. The $cond(f)$  definition boils down to the function $f$ taking $x$  as an input; proposition 1 uses $cond(\sigma_i, x)$, even though $\sigma_i$ is at an intermediate layer in the network and does not directly take as an argument. My interpretation would be that this should read $cond(f,W_iz_i)$. It doesn't really affect anything downstream as we're just taking the sup over the argument to the condition number anyways, but the notation seems a bit imprecise to me as on the LHS of the inequality we're taking the sup over calligraphic $X$ representing the input space, and on the right-hand side it's the sup over caligraphic $X_i$ representing the feature space, I feel that each intermediate feature space should have its own notation or something of the sort.
>
> We totally agree and we will modify $X$ into $X_i$, thanks for pointing this out.
>
> > In 157 the condition number for logistic sigmoid requires nonnegative features, which does not seem to be satisfied in general for intermediate neural network layers.
>
> Again, thank you. We agree. We will modify based on the following observation: If $\mathcal X_i = W_i \mathcal X_{i-1}$ then if $|z_{i-1}|\leq c_{i-1}$ entry-wise,  we have $ |x_i| \leq c_{i-1} \max_{uv} |W_i|_{uv}=:c_i $.
> Thus
> $$
> cond( \sigma ) =  \sup\_{ x \geq -c_i } |x|e^{-x}(1+e^{-x})^{-1} \leq \max\{c\_{i}, 1/e\}
> $$
>
> > In theorem 1, what is the significance of the phrase "assuming no numerical errors" --- is this referring to rounding errors, or something more serious like non-invertibility of a matrix during execution?
>
> Thanks for this question. The sentence refers to the fact that we assume that all the intermediate computed quantities (the updates and the projections) in Algorithm 2 are exact, i.e., no numerical error has been introduced.
>
> > The definition of relative error in Equation (1) doesn't sit perfectly right with me, as it is dependent on the normalization of the inputs. If all inputs were shifted by some large amount, the relative error would increase although nothing important about the data has changed. Also, the relative error at $x=0$ is always zero (or undefined?) which doesn't make a lot of sense to me. Would appreciate any clarity here.
>
> If we understand your doubt correctly, the issue you are concerned about is the fact that if we look at the condition number on a point $x$ such that $||x||$ is very large while both $||f(x)||$ and $||\delta||$ remain moderate (unif bounded for example), then the ratio $R(\tilde f,x;\delta)$ would grow as $||x||$. This is correct but it is part of what we want to measure. While we agree that if we shift the data by a large constant nothing important changes on the data while the cond number might be affected, it is also true that the sensitivity with respect to the noise should depend on the size of $x$ in the sense that we are allowed to make **relative** small perturbations: we want to look at $||\delta||/||x||\leq\epsilon$ in practice, not just $||\delta||\leq \epsilon$. If $||x||$ is large, also $||\delta||$ can in principle be chosen larger. This is what we usually do when selecting the size of the noise of the attack as we set $\epsilon = k/255$ for an RGB image, where $255=||x||_\infty$.
> As for the points where $f$ is zero: usually one discards these points as the relative error is not defined there and the absolute notion $||f(x+\delta)-f(x)||$ is used instead. This is standard in the theory of conditioning and it corresponds to the intuition that when a function is zero any variation of the function is simultaneously an absolute and a relative change.
> We hope our answer provides sufficient clarification, but we are happy to elaborate further if not.

---

> > ### Comment · Reviewer_UZkq · 2023-08-11
> >
> > Thank you for your detailed clarifications. After carefully reading other reviews / rebuttals and examining the new experiments, I believe this paper would be a good fit for NeurIPS and am raising my score to a 7.

---

### Official Review · Reviewer_sVC2 · 2023-06-13

**Soundness:** 2 fair
**Presentation:** 2 fair
**Contribution:** 2 fair
**Rating:** 5
**Confidence:** 5

**Summary:**

This paper proposes a training approach that can improve the robustness of low-rank DNN models from scratch. This method enforces the low-rank components onto the Stifel manifold and ensures well-conditioning to achieve robustness. Experimental results show that the proposed training method demonstrates robustness to the data perturbation.

**Strengths:**

+ The robustness of low-rank format DNN models is an important topic because it can improve the security of efficient DNN models.
+ Training low-rank models without initializing from a pre-trained dense model is also important to save energy costs.
+ This paper provides much theoretical stuff.

**Weaknesses:**

- The contribution is incremental. Improving the robustness of a compressed model using condition number is already proposed in [R1]. The main contribution of this paper is to re-apply it to low-rank formats. Improving the robustness of low-rank DNN models is also investigated in [R2].
- The motivation why using conditioning numbers in low-rank scenarios is unclear.
- This paper leverages Riemannian optimization to enforce the orthogonality of the left and right components, $U$ and $V$. However, this optimization algorithm contains a lot of matrix multiplications, which are extremely costly. The overall computational cost is overhead, so the proposed method is opposite to the meaning of low-rank training, which aims to save training FLOPs.
- In terms of the compression approach, why should we choose low-rankness? The overall performance of robustness and compression ratio is better than sparsity or other trends of compression techniques? This paper does not compare with any other related works like [R1] in experiments.
- The dataset and selected models in experiments are trivial. Lager datasets like ImageNet and advanced models like ResNet-50 and MobileNet should be tested.

[R1] Learning Extremely Lightweight and Robust Model with Differentiable Constraints on Sparsity and Condition Number
[R2] On the Effect of Low-Rank Weights on Adversarial Robustness of Neural Networks

**Questions:**

See Weaknesses.

---

> ### Author Rebuttal · Authors · 2023-08-08
>
> The reviewer claims that the paper is incremental because: (a) condition number as a measure of the robustness of compressed models is used already in [R1], and (b) the robustness of low-rank NNs is investigated in [R2].
> We thank the reviewer for pointing these two papers out which we had missed. However, we strongly disagree our contribution is incremental with respect to [R1] and [R2] and we strongly disagree that (quoting) "the main contribution of our paper is to re-apply [R1] to the low-rank format". **The two papers are very different** as we detail below.
>
> Re [R1]:
>
> First, [R1] does not deal with low-rank weights but with sparse weights. Second, they observe that the condition number of the layers of VGG16 grows with **unstructured** sparsification of the weights. This is aligned with our observation of exploding singular values for **structured** low-rank nets, but it does not make our observation incremental. Moreover, we notice that they do not clarify
> - which dataset they used;
> - what condition number are they measuring (product of the layers perhaps?);
> - whether they have made the same observation on other datasets/networks.
>
> Third, they propose a method based on Kronecker decomposition for *adversarial training*, where the weights are decomposed as $W = A_1\otimes A_1$. Sparsity is promoted by adding additional regularization terms and further regularization is used to promote small condition number via a tailored weight-decay combined with a logarithmic barrier on the determinant of $A_i^TA_i$.
>
> This is different than the low-rank training discussed in our paper as
> 1. we reduce the number of parameters during the entire training process by directly training matrices in low-rank factorized form; although the Kronecker format used in [R1] is low-parametric, penalty-based approaches as the ones they propose may in general make the training process more expensive than "plain training" and require tuning of many additional hyperparameters
> 1. we do not  perform any adversarial training
>
> Re [R2]:
>
> The paper takes on a different point of view, showing that adversarial training alone tends to reduce the number of parameters. Thus, using their own words, they study "whether it is possible to improve the adversarial robustness by promoting sparsity and low-rankness without adversarial training''. This second goal is more aligned with our work, however, they achieve this again by adding several penalty terms to the loss (three additional terms) which are based on the nuclear norm of the weight matrices and **do not reduce parameters during training.**
> Interestingly, they seem to observe that simultaneously low-rank and sparse weights promote robustness against adversarial examples, which goes a different way than our observation about loss of robustness when **only** low-rank constraints are imposed. However, their evaluation is limited to quite small networks and we believe it would be worth further investigating whether adding sparsity on top of low-rankness can improve robustness. Moreover, they do not propose an algorithmic way to improve robustness. Overall, we find that contribution far from ours and do not see why our approach would be incremental over theirs.
>
> We will properly reference both of these papers in the next version.
>
> Concerning your specific questions:
>
> > The motivation why using conditioning numbers in low-rank scenarios is unclear.
>
> As we discuss in §3, using the condition number is a natural way to measure the sensitivity of a NN, in general. Concerning the low-rank framework, the advantage is that computing the SVD of the small $r\times r$ matrix $S$ has a worst-case cost of $O(r^3)$ which is small when $r$ is small. Thus, the condition number of the network can be contained while training without prohibitive additional costs. This is what we state in lines 186--188. As pointed out in the response to Rev J9p9, we agree we are missing some clarification here and we will add details.
>
> > This paper leverages Riemannian optimization [...] However, this optimization algorithm contains a lot of matrix multiplications, which are extremely costly [...]
>
> We agree that the additional operations required to impose $cond(W)\leq 1 +\tau$ add computational overhead, but these are implemented on small factors only. Thus we disagree: **these operations are not extremely costly.** As also detailed in the response to Rev t9ct, the overall cost per layer of our method is $O(r(1+r)(n+m)+r^3)$ as compared to $O(nm)$ of standard training. When $r$ is small enough (e.g. $r<\sqrt n$ for $n=m$), the cost is fewer than the plain baseline.
>
> > In terms of the compression approach, why should we choose low-rankness? [..] This paper does not compare with any other related works like [R1] in experiments.
>
> Low-rank compression has the advantage that one can use the geometric properties of the matrix manifold to (a) provide theoretical analysis for the quality of the computed compressed net (see our thm 1) and (b) train directly on the manifold reducing the cost of the entire training phase. Moreover, as we point out in line 37-38, 104-105 & the response to Rev t9ct, there exists a variety of evidence showing that weights of NNs are (approximately) low-rank, which thus motivates the search of low-rank networks to start with. On the contrary, while certainly providing a number of advantages, unstructured pruning based on e.g. sparsification, does not rely on a manifold and lacks this type of supporting theory. For these reasons, we strongly believe the study of low-rank networks is a very relevant topic.
>
> Re comparison with other work, we stress that **we compare with 6 baselines**, one of which [70] uses a regularization strategy similar to [R1]
>
> > The dataset and selected models in experiments are trivial [..]
>
> We added evaluation on WideResNet/Cifar100 and we additionally tested model robustness against PGD adversarial attacks. Please see the global rebuttal block for results

---

> > ### Comment · Reviewer_sVC2 · 2023-08-19
> >
> > Thank the authors' response, which addresses my concerns. I would like to increase my original score.

---

### Official Review · Reviewer_8Zii · 2023-07-03

**Soundness:** 3 good
**Presentation:** 3 good
**Contribution:** 3 good
**Rating:** 6
**Confidence:** 4

**Summary:**

In this paper, the authors propose a low-rank training algorithm that incorporates approximate orthonormal constraints. Through a comparative analysis with previous methods for training orthonormal neural networks, the experimental results demonstrate the superiority of the proposed algorithm.

**Strengths:**

1. An algorithm was introduced to ensure that the condition number of weight matrices remains close to $1$ during the training process.

2. Comprehensive experimental results were provided in the main paper.


**Weaknesses:**

1. Some terminologies are not defined.

2. While the theoretical analysis provides some insights, it may not fully capture the advantages of the algorithm.

**Questions:**

1. Some terminologies are not pre-defined. For instance, concepts like the Stiefel manifold and tangent space may be unfamiliar to readers who have not previously explored low-dimensional spaces. Moreover, it is crucial to establish definitions for significant notations beforehand.

2. Theorem 1 establishes that the estimated matrices are closely aligned with the noisy target matrices. Nevertheless, in practical implementations, our main concern lies in the error between the estimated output and the target output. While it is possible to prove that an individual estimated weight matrix is close to its target counterpart, it does not directly guarantee that the final error won't exponentially grow as a result of multiple products and layers. Therefore, it might be worthwhile to explore how the significance of this theorem can be introduced in light of addressing the overall error in the system, considering the cumulative effects of multiple products and layers.

3. There is a missing $.$ in Eqn.(1).

---

> ### Author Rebuttal · Authors · 2023-08-08
>
> Thank you for your feedback, we provide below our response addressing the raised questions
>
> > Some terminologies are not pre-defined. For instance, concepts like the Stiefel manifold and tangent space may be unfamiliar to readers who have not previously explored low-dimensional spaces. Moreover, it is crucial to establish definitions for significant notations beforehand.
>
> Thank you for pointing this  out. We did not realize that some geometric terminology might require further  detail. We will certainly add them to the revised version.
>
> > Theorem 1 establishes that the estimated matrices are closely aligned with the noisy target matrices. Nevertheless, in practical implementations, our main concern lies in the error between the estimated output and the target output. While it is possible to prove that an individual estimated weight matrix is close to its target counterpart, it does not directly guarantee that the final error won't exponentially grow as a result of multiple products and layers. Therefore, it might be worthwhile to explore how the significance of this theorem can be introduced in light of addressing the overall error in the system, considering the cumulative effects of multiple products and layers.
>
> This is a fair point. Ideally, the Thm would provide a bound on the output of the network $f$ rather than its weights. On the other hand, while we agree that the error on $f(W)$ could blow up, even though the computed $\bar W$ is close to the target matrix $W$, this depends on the regularity of $f$. For example, for a C-Lipschitz $f$  one has $\|f(W)-f(\bar W)\|\leq C \|W-\bar W\|$. i.e., the variation of $f$ is controlled.
> Interestingly, one could obtain a relative version of such statement by looking into the condition number of $f$ with respect to the weights rather than the data and it would directly follow from Eq.(1)  that,
> $$
>     \frac{\|f(W)-f(\bar W)\|}{\|f(W)\|}\leq cond_W(f)\frac{\|W-\bar W\|}{\|W\|} .
> $$
> A similar bound as the one in Prop1 holds for $cond_W$. We can add this type of analysis/discussion to the paper if the reviewer considers this would improve the quality of the contribution.

---

> > ### Comment · Reviewer_8Zii · 2023-08-11
> >
> > Thank you for your rebuttal. I will maintain my current score.

---

### Official Review · Reviewer_J9p9 · 2023-07-04

**Soundness:** 3 good
**Presentation:** 3 good
**Contribution:** 3 good
**Rating:** 7
**Confidence:** 4

**Summary:**

Some work has used low-rank matrix factorizations in order to reduce training and inference costs. However, the authors demonstrate that these low-rank models are actually more brittle, i.e., susceptible to adversarial perturbations -- they tend to have large/exploding singular values. Consequently, the authors propose a technique to train low-rank networks while constraining the condition number, alleviating this problem.

**Strengths:**

- The paper triangulates a well-defined problem and proposes an effective solution -- the paper was generally quite clear and well-motivated
- Demonstrates the method works in practical experience and also via an approximation theorem
- Investigation of exploding singular values in low-rank networks
- Comparisons for a variety of compression ratios / perturbation magnitudes
- Good explanation of background and related work


**Weaknesses:**

- Would be good to substantiate "improving the network robustness without affecting training nor inference costs" -- it's not obvious this is true to me since the method involves, e.g., projection onto the Stiefel manifold
- A conclusion would be nice -- I think you could save space by removing the definition of robust accuracy, which readers probably know

**Questions:**

This is minor, but why did you only produce perturbations with FGSM as opposed to, e.g., PGD? I'd be curious how your compressed models withstand somewhat more powerful adversaries.

**Limitations:**

The authors have adequately addressed limitations, e.g., noting that they introduce a new hyperparameter that may have to be tuned.

---

> ### Author Rebuttal · Authors · 2023-08-08
>
> Thank you very much for your encouraging feedback. Below we provide our response to your weak points and questions.
>
> >  Would be good to substantiate "improving the network robustness without affecting training nor inference costs" -- it's not obvious this is true to me since the method involves, e.g., projection onto the Stiefel manifold
>
> Thanks for this point - we agree that it is not obvious that the cost is not significantly affected and we will clarify this in the next iteration of the paper. As detailed in the response to Reviewer t9ct, we will add details about the computational cost of the algorithm. Specifically, the particular sentence at line 188 "improving the network robustness without affecting training nor inference costs" refers to a comparison with training using just the low-rank factors without the Stiefel manifold projection. Note that in our setting, the projection is done for the matrix $S$ which is $r\times r$ so the cost of projecting has a worst-case complexity of $O(r^3)$. When $r\ll n,m$ this cost is negligible and in this sense it does not affect training nor inference costs. We will support this sentence at line 188 with details.
>
> > A conclusion would be nice -- I think you could save space by removing the definition of robust accuracy, which readers probably know
>
>  Thanks, we will definitely add a conclusion, as also indicated in the response to Reviewer t9ct
>
> > This is minor, but why did you only produce perturbations with FGSM as opposed to, e.g., PGD? I'd be curious how your compressed models withstand somewhat more powerful adversaries.
>
> We have added results for PGD and also extended the evaluation to WideResNet on Cifar10 and Cifar100

---

### Official Review · Reviewer_t9ct · 2023-07-06

**Soundness:** 3 good
**Presentation:** 2 fair
**Contribution:** 3 good
**Rating:** 6
**Confidence:** 4

**Summary:**

A brief summary of the paper is as follows:
- Motivation: Make deep neural networks more efficient while maintaining robustness to adversarial attacks.
- The authors propose a combined low-rank and approximate orthonormalisation constraint to help simultaneous model conditioning and compression during training.
- The results show that both accuracy and adversarial robustness remain high.

It is well written and tackles an important problem, however it is missing references and comparisons to key existing works.

**Strengths:**

The main strength of this paper lies in the writing and the novelty of the approach. Optimizing the low-rank decompositions alongside an orthonormal constraint is a very interesting approach that deserves an in-depth study. Preventing explosion of singular values is important and an interesting research direction that will be of value beyond low-rank neural networks and adversarial attacks. After addressing the weaknesses of this paper, the strengths can make this paper into a very complete piece of work.

**Weaknesses:**

- Paper does not mention previous studies that approach this same problem: adversarial robustness with low-rank constraints. [1] introduces tensor dropout, a method of increasing robustness (including robustness to adversarial attacks) using a stochastic dropout mechanism in the learned weight decompositions. It would be good to get a comparison to that baseline or at the very least a discussion of the similarities and differences between the approaches.
- More discussion of the results and method would be good. There a number of questions that arise with this work that remain unanswered. Even though answering all of them might be out of scope for a single paper, it would be worth having the authors discuss more aspects of the approach they introduce here. I refer to my list of questions below, all of which would be interesting discussion points where the pros and cons of different decisions can be weighted.
- The paper is missing a well-defined conclusion. That means the paper ends abruptly without comment on the impact of this work and what the authors see as possible next steps.

[1] Kolbeinsson, A., Kossaifi, J., Panagakis, Y., Bulat, A., Anandkumar, A., Tzoulaki, I., & Matthews, P. M. (2021). Tensor dropout for robust learning. IEEE Journal of Selected Topics in Signal Processing, 15(3), 630-640.

**Questions:**

These are questions that I have for the authors but I highly recommend they consider expanding the discussion and limitations part of their paper. They could touch on these points.
- In the end of the abstract it is stated that "This is shown by extensive numerical evidence and by our main approximation theorem that shows the computed robust low-rank network well-approximates the ideal full model, provided a highly performing low-rank sub-network exists." My question is: is it possible to check if a highly performing subnetwork exists, and does the orthonormal constraint affect the group of available subnetworks?
- Related, how is the r parameter selected? Any suggestions for practice?
- “Works only for wide enough networks” How wide is enough?
- What is the added computational complexity of the new optimization? What is the time/memory cost relationship between the low-rank and the full network?
- Why are VGG16 tested instead of the more common Resnets?
- What attack is used for the adversarial experiments?
- How sensitive is the method to choice of r?

**Limitations:**

The paper has a brief discussion on the limitations but this should be expanded to greatly improve the paper. Getting more candid explanations and thoughts on the limitations from the authors themselves would be welcomed. Possible discussion points would be the  sensitivity to parameter r (related to compression ratio). This is a trade off between performance and computation and can be different for different applications, however it is important to know the scope of available r values. Another very interesting limitation is the additional time cost of this method, specifically in relation to the additional optimization steps that need to be done. What is the relative added computation of this? A discussion of what conditions are needed for this optimization is of value would be very welcome.

---

> ### Author Rebuttal · Authors · 2023-08-08
>
> Thank you very much for your useful feedback.
> Below we provide a detailed response to all the raised criticism. We notice that all raised weaknesses, except for one point concerning a missing comparison/reference with paper [1], are entirely related to the missing concluding section and some missing discussion of the results which (quoting) "would be worth having".
> **We are quite surprised these seemingly minor weak points correspond to a borderline rejection evaluation and would really appreciate additional clarification and feedback from the reviewer on what should be improved.**
> As highlighted below, we will complement the existing discussion with details re the missing points and we will add a concluding section.
>
> Re the specific questions:
>
> > Comparison with [1]
>
> We provide comparison in the global rebuttal
>
> >  In the end of the abstract it is stated that [...] is it possible to check if a highly performing subnetwork exists, and does the orthonormal constraint affect the group of available subnetworks?
>
> This is a very good point. Proving the existence of highly performing low-rank subnets is a challenging problem that is currently subject to intensive investigation in the literature. **As we state at the end of §2 (lines 104-106), there is a variety of (quoting) "recent work that shows the existence of high-performing low-rank nets in e.g. deep linear models [7,16,23,45]"**.  The extension to more general classes of NNs (which include nonlinearities) is of course very important as well as more challenging and still largely open to our knowledge.   We refer for instance to the recent work [Galanti et al, SGD and Weight Decay Provably Induce a Low-Rank Bias in Deep Neural Networks] for recent insights in this direction.
>
> > Related, how is the r parameter selected? Any suggestions for practice? **and** How sensitive is the method to choice of r?
>
> The parameter $r$ is responsible for the compression rate of the method and thus its selection depends also on the memory and the computational budget provided.  As discussed at lines 242-246,  the number of parameters (per layer) used by the proposed model is $r(n+m+r)$ and the compression rate depends on $r$ as
> $$
> cr = \frac{\text{parameters full net}-\text{parameters compressed net}}{\text{parameters full net}}=1-\frac{r(n+m+r)}{nm}.
> $$
> One can use these formulas to select $r$, given the available resources.
> On the other hand, we agree that – connecting to the previous question – a theoretical analysis providing insight on the choice of $r$ based on the investigation of the rank properties of the full network would be ideal. However, this is a challenging question that is the subject of a broad effort by the community and has no clear answer yet.
> Finally, we also mention that the parameter $r$ can be fine-tuned algorithmically and that there are some papers that address this point (e.g. [51,70] from our paper) however, the integration of these hyperparameter optimization steps into our proposed framework is not straightforward and is left to future investigations.
> As we highlight in the limitations section, while this is a limit of the proposed analysis, our current evaluation provides enough details to hint at the way the performance changes with $r$, as we show results for both moderate (50\%) and intensive (80\%) compression rate regimes, showing consistent performance.
>
> > "Works only for wide enough networks" How wide is enough?
>
> Thanks for this question. Our point here was simply that, given the compression rate formula $cr = 1 -\frac{r(n+m+r)}{nm},$ if the layer width parameters $m$ and $n$ are not large enough, the method would not perform compression. This is simply due to the fact that the numerator $r(n+m+r)$ is much smaller than the denominator only if $n,m\gg r$. For example, if $n=m$, then we have a positive compression rate $cr>0$ only if $n>2.5*r$. We will adjust the text here to clarify this point in the revised version of the paper.
>
> > What is the added computational complexity of the new optimization? What is the time/memory cost relationship between the low-rank and the full network?
>
> Thank you for pointing this missing information out. We detail the computational complexity of the method below and will add a paragraph with this info in the new version.
>
> Each pass of Alg1 is done against a batch $x_{\mathrm{batch}}$. In order to obtain minimal computational costs, in the algorithm we evaluate $USV^\top x_{\mathrm{batch}}$ sequentially: first $v=V^\top x_{\mathrm{batch}}$, then $u=Sv$, finally $Uu$. Assuming the size of the batch is negligible with respect to $n$ and $m$, the cost of these steps is $O(rm), O(r), O(rn)$, respectively. Adding the bias and evaluating the activation function requires $O(n)$ operations. Hence, overall we have a cost per layer of $O(r(n+m+1))$. Taping the forward evaluation to compute the gradient with respect to $U,S,V$ does not affect these costs as this is also done sequentially against the chosen batchsize. The QR decompositions used for $U$ and $V$ require $O(r^2n)$ and $O(r^2m)$ operations respectively, $O(r^2(n+m))$ overall. Finally, computing the SVD in the projection step for $S$ requires a worst-case cost of $O(r^3)$. Hence the overall cost per layer and per step is $O(r(1+r)(n+m)+r^3)$ as opposed to dense training, which requires $O(nm)$ operations. If $r\ll n,m$ then the method is cheaper than the full baseline. For example, if $n=m$, this happens provided $r<\sqrt n$.
>
> > Why are VGG16 tested instead of the more common Resnets?
>
> Our academic computational power did not allow us to scale to very large networks/datasets at the time of submission. We have now tested on Wideresnet on Cifar10 and Cifar100, **new results are in the global rebuttal**
>
> > What attack is used for the adversarial experiments?
>
> Our experiments are done using the FSGM (see lines 298-299 in the main paper). We have now added results for PGD attack. **new results are in the global rebuttal**

---

> > ### Comment · Reviewer_t9ct · 2023-08-18
> > **Reply to authors**
> >
> > I thank the reviewers for their response.
> >
> > Similar to them, I am surprised to see that they consider a missing conclusion, missing references and missing details on complexity to be minor points. I consider them quite important, hence my score.
> >
> > I also believe the authors misinterpreted the reason I highlighted the missing references. That is likely due to my comment not being clear enough. The work presented here and reference [1] in my previous comment are not methodologically very similar, however it is important to acknowledge related work (even if the approach is not comparable) when making statements such as "However, their [low-rank weights parametrization models'] robustness with respect to adversarial perturbations has been largely unexplored so far." is not representative without full acknowledgement. To be very clear, the methodology of this work and that in [1] do not overlap so that it raises any special concerns over novelty. They simply exist in the same domain: using low-rank networks to improve adversarial robustness.
> >
> > With the conclusion added, complete references for related work and the seven loose ends that I questioned in my review all amended, I am prepared to upgrade my score.

---

> > > ### Author Response · Authors · 2023-08-20
> > >
> > > Dear reviewer,
> > >
> > > Thank you very much for your positive feedback and for the clarifications.
> > > We agree with your point of view and will make the necessary changes to the manuscript.

---

### Author Rebuttal · Authors · 2023-08-09

We thank all the reviewers for their valuable feedback and useful comments and criticism. We provide a detailed response to all weak points and questions in the individual rebuttal blocks below. We use this general rebuttal block to report additional evaluation in response to a number of reviewers' questions.

# Comparison with reference [1] from Rev t9ct

[1] proposes a singular value dropout approach where a number of singular values are removed at random during training. However, they only implement this form of randomized rank reduction in the final tensor regression layer. This is not comparable with our approach which instead reduces the rank of the entire network (all the layers). We implemented the approach of [1] on the entire network (instead of just the final layer as it is done in [1]), however the results are not competitive and we believe they are not worth being included in the paper. We will instead reference [1] in the related work section.

The tables 4 and 5 in the attached pdf show the performance of [1] when the dropout is performed on the entire network at different compression ratios. A similar performance is observed for zero compression rate.

# Additional evaluation on WideResNet and using PGD adversarial attack
In table 1, we provide robustness results using PGD attack with 10 steps and L2 norm. (This table is analogous to Tab.3 in supplementary material, just with PGD10-L2 instead of FGSM). As demonstrated, our model with 50% compression rate still outperforms baseline,  Riemannian methods and other low-rank methods.

Additionally we implemented CondLR for WideResnet 16-4(depth-width) and trained it on CIFAR10 and CIFAR100.
Table 2 (CIFAR10) provides comparison for two settings of our method ($\tau = 0.0$, $\tau = 0.1$) against baseline in the case of compression rates 50% and 80%. Performance of each model is measured in terms of the robustness against FGSM and PGD10 attacks with L$\infty$-norm. As demonstrated, for 50% compression rate our method gets better results than baseline model and 80% compression rate scores reasonably close to baseline.

Table 3 (CIFAR100) provides comparison for two settings of our method ($\tau = 0.1$, $\tau = 0.5$) against baseline for compression rate 50% (compared with FGSM and PGD10 attacks with L$\infty$-norm). Similarly, better performance of our method against baseline is observed.

---

### Decision · Program_Chairs · 2023-09-21

**Decision:**

Accept (poster)

**Comment:**

This work focuses on the study of low-rank training. The authors demonstrate that conventional low-rank training methods suffer from poor adversarial robustness, primarily attributable to the ill-conditioning of weight matrices during training. The authors tackle this challenge by imposing constraints on low-rank weight matrices to make them approximately orthogonal.

The authors have successfully addressed the concerns raised by the reviewers, and all reviewers recommend acceptance. I concur with the reviewers and also recommend acceptance. The authors are advised to incorporate the discussion of comparison with existing research and include additional experimental results in the final version.